# TIES-MERGING: Resolving Interference When Merging Models

**Prateek Yadav**[1]   **Derek Tam**[1]
**Leshem Choshen**[2,3]   **Colin Raffel**[1]   **Mohit Bansal**[1]
[1] University of North Carolina at Chapel Hill   [2] IBM Research   [3] MIT
leshem.choshen@ibm.com
{praty,dtredsox,craffel,mbansal}@cs.unc.edu

## Abstract

Transfer learning – i.e., further fine-tuning a pre-trained model on a downstream task – can confer significant advantages, including improved downstream performance, faster convergence, and better sample efficiency. These advantages have led to a proliferation of task-specific fine-tuned models, which typically can only perform a single task and do not benefit from one another. Recently, model merging techniques have emerged as a solution to combine multiple task-specific models into a single multitask model without performing additional training. However, existing merging methods often ignore the interference between parameters of different models, resulting in large performance drops when merging multiple models. In this paper, we demonstrate that prior merging techniques inadvertently lose valuable information due to two major sources of interference: (a) interference due to redundant parameter values and (b) disagreement on the sign of a given parameter's values across models. To address this, we propose our method, TRIM, ELECT SIGN & MERGE (TIES-MERGING), which introduces three novel steps when merging models: (1) resetting parameters that only changed a small amount during fine-tuning, (2) resolving sign conflicts, and (3) merging only the parameters that are in alignment with the final agreed-upon sign. We find that TIES-MERGING outperforms several existing methods in diverse settings covering a range of modalities, domains, number of tasks, model sizes, architectures, and fine-tuning settings. We further analyze the impact of different types of interference on model parameters, and highlight the importance of resolving sign interference.[1]

## 1   Introduction

Pre-trained models (PTMs) have become widespread in many real-world applications [91, 6]. Using PTMs typically involves fine-tuning them to specialize on a specific task [69, 12], which can lead to improved performance with less task-specific labeled data. These benefits have resulted in the release of thousands of finetuned checkpoints [81] derived from popular PTMs such as ViT [14] for vision and T5 [58] for language. However, having a separate fine-tuned model for each task has various drawbacks: (1) for each new application, a separate model has to be stored and deployed [17, 89], and (2) models trained in isolation cannot leverage information from related tasks to improve in-domain performance or out-of-domain generalization [66, 58, 75]. Multitask learning [66, 57] could address these concerns but requires costly training and simultaneous access to all tasks [17]. Moreover, it can be complex and resource-intensive to determine how best to mix datasets to ensure that multitask training is beneficial for all tasks [55, 54, 80, 52, 2, 17].

---

[1]Our code is available at https://github.com/prateeky2806/ties-merging

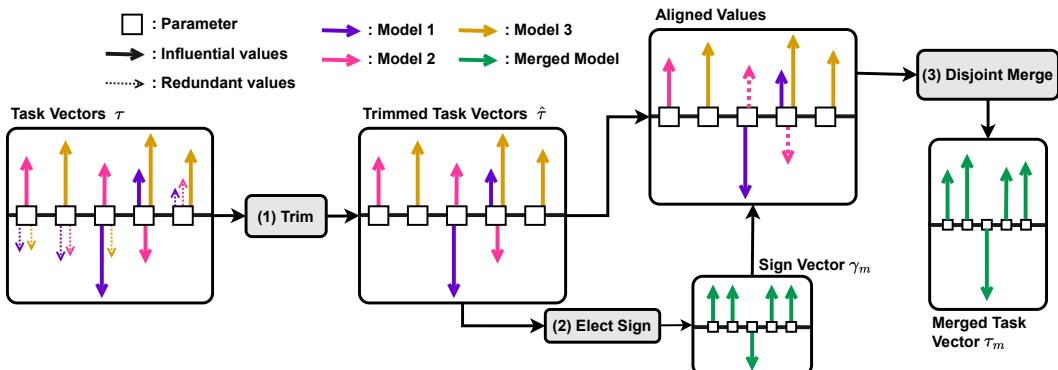

Figure 1: A depiction of the steps involved in TIES-MERGING. We visualize each parameter in a model as a square. The arrows depict the update (task vector, $\tau$) to a parameter produced by fine-tuning on different tasks (coded by colors), with direction denoting sign and length denoting magnitude. We first *trim* the task vector values based on their magnitude, then we *elect* the sign for each parameter ($\gamma_m$, green vector containing $+1$ or $-1$) by resolving sign conflicts. Finally, we pick only the values that align with the elected sign and take their mean as the final parameter value.

Recently, a growing body of research has focused on *model merging* [40]. One application of merging involves combining multiple task-specific models into a single multitask model without performing additional training. Previous works merge models by summing the individual model weights with different weighting schemes, either via a simple average [9, 28, 83], via more sophisticated means that incorporate parameter importance [45] or account for permutation invariances [1, 31, 70, 74, 42]. Combining fine-tuned models in this way can be seen as adding together *task vectors* [29] that are computed by subtracting the pre-trained model's parameter values from those of the fine-tuned model.

While weighted averaging of model parameters has proven effective for merging, all of these methods ignore the possibility that values may interfere across models, thereby harming the performance of the merged model. In this paper, we first demonstrate that interference can stem from two major causes (see Fig. 2), both of which can reduce parameter magnitudes in the merged model and eliminate subtle distinctions among values: (1) INTERFERENCE FROM REDUNDANT PARAMETERS: Previous studies on model pruning [25, 76] have shown that during fine-tuning, many model parameters can change over the course of fine-tuning [63] but only have a small impact on performance. However, when merging a parameter that is influential for one model but redundant (i.e. not influential) for other models, the influential value may be obscured by the redundant values, lowering the overall model performance ($\bigcirc$ in

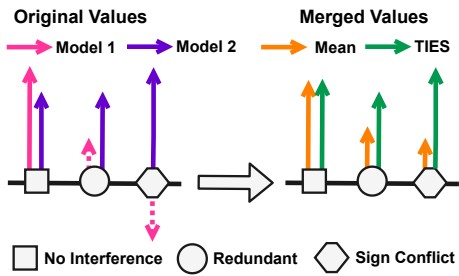

Figure 2: Different types of conflict and merged outputs produced by either averaging or TIES-MERGING. The parameters causing interference are denoted by dotted arrows.

Fig. 2). (2) INTERFERENCE FROM SIGN DISAGREEMENT: A given parameter might have a positive value for some models and a negative value for others. Consequently, employing simple averaging might compromise the performance on both tasks ($\bigcirc$ in Fig. 2). In both of these situations, simply aggregating the values lead to interference that shrinks the parameter's value in the merged model. This interference between influential parameters might explain why the performance gap between the merged model and multitask-trained model increases as the number of models increases [31].

To address these sources of interference, we propose TIES-MERGING (TRIM, ELECT SIGN & MERGE) method, a method for merging models by combining task vectors that has three steps (visualized in Fig. 1): First, we trim each task vector to retain only the influential parameter values by setting the redundant values in each task vector to zero (or, equivalently, resetting the fine-tuned

parameter value back to the value from the pre-trained model). After this step, sign conflicts may still persist among influential parameter values, as visualized in Fig. 4. Our second step therefore resolves the sign conflicts between different values and our last step only averages parameters whose sign agrees with the direction of the largest total movement across models.

We demonstrate the effectiveness of our proposed TIES-MERGING method in various setups with: (1) different modalities, including language and vision benchmarks, (2) distinct model sizes and families, such as T5-base and T5-large [58] as well as ViT-B/32 and ViT-L/14 [14], (3) in-domain and out-of-domain tasks, (4) full finetuning or parameter-efficient finetuning, and (5) in the presence or absence of a validation set for setting merging hyperparameters. We show that TIES-MERGING outperforms other merging methods, such as Task Arithmetic [29], RegMean [31], Fisher Merging [45], and weight averaging [9, 82] across all these experimental settings. Notably, for in-domain evaluation, TIES-MERGING outperforms the strongest baseline by an average of 2.3% and 1.7% absolute in NLP and vision settings (Table 1), respectively. For out-of-domain generalization (Table 2), TIES-MERGING outperforms the strongest baseline by 1.0% and 4.4% absolute for T5-base and T5-large models respectively. In Section 7, we perform ablations on our method components and demonstrate the impact of interference on parameter values. Additionally, we showcase the increased advantage of TIES-MERGING over task arithmetic [29] as the number of tasks increases. Finally, we examine the importance of obtaining the correct sign vector. Our results and analysis establish TIES-MERGING as a powerful and effective method for combining fine-tuned models into a single multi-task model.

## 2 Related Work

**Loss Landscape and Weight Interpolation.** While the loss function of a neural network is generally non-convex, recent work has demonstrated that the parameter values from different training runs can sometimes be interpolated without increasing the loss (i.e. they are *mode-connected*) [15, 20, 21, 32, 22]. For example, Frankle et al. [19] showed that if a part of the optimization trajectory is shared between two neural networks then they can be interpolated without lowering accuracy. On the other hand, Neyshabur et al. [48] showed that naively interpolating two neural networks with completely disjoint optimization trajectories can result in a catastrophic drop in their accuracies. Entezari et al. [16] hypothesized that if we account for the permutation symmetry of neural networks, then all neural networks of a given architecture trained on the same dataset are linear mode connected. Ainsworth et al. [1], Singh and Jaggi [70], Wang et al. [79] therefore used techniques based on finding permutations [79, 1] and optimal transport [70] to better align neural networks trained from scratch so that they can be interpolated without increasing the loss.

**Model Merging and Different Use Cases.** Different fine-tuned models initialized from the same pre-trained model effectively share a part of the optimization trajectory, and can therefore often be merged without accounting for permutation symmetry [82, 83, 29, 31]. Therefore, merging fine-tuned models can improve performance on a single target task [30, 23, 82, 9], improving out-of-domain generalization [31, 29, 7, 4, 60, 59], creating a multitask models from different tasks [31, 29, 38], for federated learning [46, 41], compression [39], multimodal merging models [72], continual learning [86, 85], and other settings [38, 13]. The range of applications has led to a proliferation of methods to improve beyond simple parameter averaging. *RegMean* [31] proposed a closed-form solution for the merged model's parameters by solving a local linear regression problem for each individual linear layer in the model. However, this requires transmitting additional data statistics that are the same size as the model and requires additional inference steps to calculate them. *Fisher Merging* [45] goes beyond simple averaging to identify the importance of individual parameters using Fisher Information Matrix [18, 3, 34] and uses it to weigh the parameters in each model when merging. However, this shows little gains when merging multiple checkpoints and also requires computing gradients which has a high memory cost. *Task Arithmetic* [29] presented a method for merging models by generating task vectors and performing arithmetic operations, such as addition, to obtain a multitask checkpoint. A concurrent work by Ortiz-Jiménez et al. [51] provided theoretical insights on model merging based on the weight disentanglement property that arises during pretraining. They showed that finetuning models in their tangent space enhance this property, leading to better-merged models. Our method follows these past works on model merging but additionally takes into account the interference between different parameters during merging.

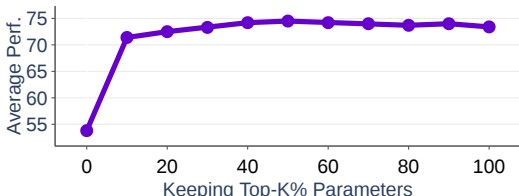
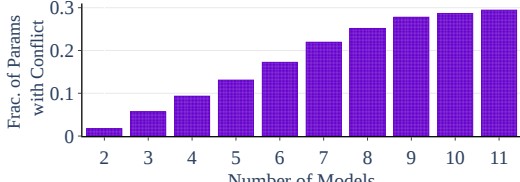

Figure 3: **Performance depends on a small fraction of high-magnitude parameters.** For each task vector, we keep only the largest - top-$k\%$ parameters and plot the average performance across eleven tasks. Keeping only the top-20% of the parameter does not degrade the performance.

Figure 4: **Sign conflicts occur even after trimming and increase with the number of models.** We plot the fraction of parameters that have a sign conflict after trimming versus the number of models being merged.

## 3 Background and Motivation

**Problem Setting** Given a set of tasks $\{t_1, \ldots, t_n\}$ and a pre-trained model such as T5 [58] or ViT [14], we either finetune the entire model or employ a parameter-efficient finetuning (PEFT) method [43, 26]. In both cases, we denote the trainable parameters as $\theta$, the initialization as $\theta_{\text{init}}$, and the finetuned parameters as $\theta_{\text{ft}}$. In this paper, we assume access to finetuned model parameters $\theta_{\text{ft}}$ for multiple tasks and devise a method to merge the weights of these models into a single multitask model proficient on both in-domain and out-of-domain datasets. We follow Ilharco et al. [29] and perform merging with task vectors. Specifically, for a task $t$, the task-vector $\tau_t \in \mathbb{R}^d$ is defined as $\tau_t = \theta_{\text{ft}}^t - \theta_{\text{init}}^t$. This operation allows us to focus on the changes that happen during the fine-tuning phase of each task-specific model and is equivalent to computing a weighted average of the models' weights with appropriate scaling.

**Redundancies in Model Parameters.** First, we demonstrate that in a given task vector, many values are redundant (denoted by ○ in Fig. 2), and removing them does not affect the performance of the task. Specifically, Fig. 3 shows the average performance across eleven task-specific models when *"trimming"* each task vector to retain only the top-$k\%$ largest-magnitude values and resetting the rest to their initial value (i.e. setting the corresponding value in the task vector to 0). Fig. 3 shows the average performance across varying values of $k$, demonstrating that keeping only the top-20% of values delivers comparable results to retaining all parameters. For additional details and the results on the T5 model, please refer to Appendix C.3. This shows that many parameter changes introduced during fine-tuning are redundant. Hence, disregarding those values during merging could prevent interference with the influential parameters without compromising the task's performance.

**Algorithm 1** TIES-MERGING Procedure.

**Input:** Fine-tuned models $\{\theta_t\}_{t=1}^n$, Initialization $\theta_{\text{init}}$, $k$, and $\lambda$.
**Output:** Merged Model $\theta_m$
**forall** $t$ **in** $1, \ldots, n$ **do**
   ▷ Create task vectors.
   $\tau_t = \theta_t - \theta_{\text{init}}$
   ▷ Step 1: Trim redundant parameters.
   $\hat{\tau}_t \leftarrow \text{keep\_topk\_reset\_rest\_to\_zero}(\tau_t, k)$
   $\hat{\gamma}_t \leftarrow sgn(\hat{\tau}_t)$
   $\hat{\mu}_t \leftarrow |\hat{\tau}_t|$
**end**
▷ Step 2: Elect Final Signs.
$\gamma_m = sgn(\sum_{t=1}^n \hat{\tau}_t)$
▷ Step 3: Disjoint Merge.
**forall** $p$ **in** $1, \ldots, d$ **do**
   $\mathcal{A}^p = \{t \in [n] \mid \hat{\gamma}_t^p = \gamma_m^p\}$
   $\tau_m^p = \frac{1}{|\mathcal{A}^p|} \sum_{t \in \mathcal{A}^p} \hat{\tau}_t^p$
**end**
▷ Obtain merged checkpoint
$\theta_m \leftarrow \theta_{\text{init}} + \lambda * \tau_m$
**return** $\theta_m$

**Disagreement between Parameter Signs:** Different fine-tuned models might introduce opposing changes to a parameter in their task vectors, causing interference due to conflicting signs (denoted by ○ in Fig. 2). Fig. 4 presents an analysis of the frequency of sign conflicts when merging varying numbers of models. We first trim the task vectors for eleven tasks by keeping only the top 20% of influential parameters. Then, we plot the percentage of parameters that have a sign conflict as we

increase the number of models to be merged from 2 to 11. Notably, sign conflicts occur even when merging only 2 models from different tasks or when merging multiple models from the same task (see Appendix Figure 10), and the likelihood of a sign conflict increases with the number of models being merged. For additional details and the results on the T5 model, please refer to Appendix C.3.

## 4  TIES-MERGING: TRIM, ELECT SIGN & MERGE

To address the aforementioned issues, we present TIES-MERGING (TRIM, ELECT SIGN & MERGE), which aims to address the kinds of interference mentioned above before performing merging.

### 4.1  Preliminaries

A task vector $\tau_t \in \mathbb{R}^d$ represents a direction and the amount of movement required in the $d$-dimensional parameter space relative to the initialization that leads to a low loss region for the task $t$. Each entry in $\tau_t$ (corresponding to a particular parameter) can be thought of as an axis in the $d$-dimensional space. The sign of a parameter denotes the direction along this axis (positive or negative) that decreases the loss on task $t$. Hence, a given task-vector $\tau_t$ can be decomposed into a *sign vector* $\gamma_t \in \mathbb{R}^d$ and a *magnitude vector* $\mu_t \in \mathbb{R}^d$ as $\tau_t = \gamma_t \odot \mu_t$, where $\odot$ is the elementwise product. Formally, $\gamma_t = \text{sgn}(\tau_t)$, where $\text{sgn}(x) * |x| = x$ and returns a value of $+1$, $0$, or $-1$. The magnitude vector $\mu_t$ is defined as $\mu_t = |\tau_t|$ and the value $\mu_t^i$ tells us the movement required in the $i$-th dimension from the initialization.

### 4.2  Steps in TIES-MERGING

To merge multiple task-specific models $\{\theta_t\}_{t=1}^n$, we first create corresponding task vectors $\{\tau_t\}_{t=1}^n$. Given these task vectors, TIES-MERGING method follows three steps in order to perform a merge (see Fig. 1 for a diagram and Algorithm 1):

1. **Trim:** For each task $t$, we trim the redundant parameters from the task vector $\tau_t$ to create $\hat{\tau}_t$ by keeping the top-$k\%$ values according to their magnitude and trimming the bottom $(100-k)\%$ of the redundant parameters by resetting them to 0. This can be decomposed further as $\hat{\tau}_t = \hat{\gamma}_t \odot \hat{\mu}_t$.

2. **Elect:** Next, we create an aggregate elected sign vector $\gamma_m$ for the merged model that resolves the disagreements in the sign for each parameter $p$ across different models. To create the elected sign vector, we choose the sign with the highest total magnitude across all relevant models. For each parameter $p \in \{1, 2, \ldots, d\}$, we separate the values $\{\hat{\tau}_t^p\}_{t=1}^n$ based on their sign ($+1$ or $-1$) and take their sum to calculate the total mass (i.e., total magnitude) in the positive and the negative direction. We then assign $\gamma_m^p$ as the sign with greater total movement. This can be efficiently computed using $\gamma_m^p = \text{sgn}(\sum_{t=1}^n \hat{\tau}_t^p)$.

3. **Disjoint Merge:** Then, for each parameter $p$, we compute a *disjoint mean* by only keeping the parameter values from the models whose signs are the same as the aggregated elected sign and calculate their mean. Formally, let $\mathcal{A}^p = \{t \in [n] \mid \hat{\gamma}_t^p = \gamma_m^p\}$, then $\tau_m^p = \frac{1}{|\mathcal{A}^p|} \sum_{t \in \mathcal{A}^p} \hat{\tau}_t^p$. Note that the disjoint mean always ignores the zero values.

Given the final merged task vector $\tau_m$, we scale it and add it to the initial parameter values to obtain the merged model parameters $\theta_m$ as $\theta_m = \theta_{\text{init}} + \lambda * \tau_m$, where $\lambda$ is a scaling hyperparameter (as used in past work [29]).

## 5  Experimental Setup

**Baseline Methods.**   We compare TIES-MERGING with four baseline merging methods: (1) **Simple Averaging** [9, 82] calculates the element-wise mean of all the individual models and can be expressed as $\theta_m = \sum_{t=1}^n \theta_t / n$. (2) **Fisher Merging** [45] uses a diagonal approximation of the Fisher Information Matrix $\hat{F}_t$ [34, 3, 18] to measure the importance of each parameter for task $t$, where $\hat{F}_t = \mathbb{E}_{x \sim D_t} \mathbb{E}_{y \sim p_{\theta_t}(y|x)} \nabla_{\theta_t} (\log p_{\theta_t}(y|x_t))^2$. The final merged model is obtained by reweighting each parameter in each fine-tuned model by the corresponding value in the model's approximate Fisher matrix as $\theta_m = \sum_{t=1}^n \hat{F}_t \theta_t / \sum_{t=1}^n \hat{F}_t$. (3) **RegMean** [31] computes a closed-form solution to

| Method (↓) | Validation | PEFT | Full Finetuning | | | |
|---|---|---|---|---|---|---|
| Model (→) | | $(IA)^3$ | T5-Base | T5-Large | ViT-B/32 | ViT-L/14 |
| FINE-TUNED | - | 71.4 | 82.8 | 88.8 | 90.5 | 94.2 |
| MULTITASK | - | 73.1 | 83.6 | 88.1 | 88.9 | 93.5 |
| AVERAGING [82, 9] | ✗ | - | 65.9 | 59.6 | 65.8 | 79.6 |
| TASK ARITHMETIC [29] | ✗ | - | **73.2** | 73.5 | 60.4 | 83.3 |
| **TIES-MERGING** | ✗ | - | 69.7 [-3.2] | **74.4** [+0.9] | **72.4** [+6.6] | **86.0** [+2.7] |
| FISHER MERGING [45] | ✓ | 62.2 | 68.9 | 64.6 | 68.3 | 82.2 |
| REGMEAN [31] | ✓ | 58.0 | 71.2 | 73.2 | 71.8 | 83.7 |
| TASK ARITHMETIC [29] | ✓ | 63.9 | 73.2 | 73.3 | 70.1 | 84.5 |
| **TIES-MERGING** | ✓ | **66.4** [+2.5] | **73.9** [+0.7] | **76.9** [+3.6] | **73.6** [+1.8] | **86.0** [+1.5] |

Table 1: Comparing model merging methods across multiple fine-tuning settings and modalities (NLP and Vision) with and without the availability of a validation set.

a least-squares regression problem that aims to minimize the distance between the merged model's activations and the individual models' activations as $\theta_m = (\sum_{t=1}^{n} X_t^T X_t)^{-1} \sum_{t=1}^{n} (X_t^T X_t \theta_t)$, where $X_t$ is the input activation of a given layer. (4) **Task Aritithmetic** [29] scales and then adds the task vectors to the initial model to produce the merged model as $\theta_m = \theta_{init} + \lambda * \sum_{t=1}^{n} \tau_t$. In addition to these baselines, we present the performance of the individual **fine-tuned models** involved in the merging process as well as the performance of a **multi-task model** trained on the concatenation of all tasks' datasets. For more details on compute resources, dataset licenses, and the finetuning procedures, refer to Appendix C.1, C.2, and C.6.

**Merging in Absence of the Validation Set.** Prior works [29, 45, 82] on model merging assume access to a validation set, which is utilized to compute the Fisher matrix or tune hyper-parameters. To avoid the need for a validation set, RegMean [31] proposed storing and transmitting inner product matrices of the training data for each task that are the same size as the original model. This can quickly become expensive for large models as the storage and transmission scale linearly with model size and the number of tasks.

To consider the setting where no validation set is available, we developed a generic recipe of TIES-MERGING with fixed hyperparameters that could be applied in any setting without hyperparameter tuning on a validation set. The recipe keeps the top-20% parameters in the task vector resetting the rest to 0 and sets $\lambda = 1$. We chose this recipe based on results in the parameter-efficient fine-tuning (PEFT) setting, so we only apply it to the unseen settings of full model fine-tuning on ViT (vision) and T5 (language) models. We also compare TIES-MERGING with the Task Arithmetic method without a validation set by utilizing the recommended value of $\lambda = 0.4$ [29]. For further details on how this recipe was created please refer to Appendix C.4.

# 6 Main Results

Our main goal is to merge multiple task-specific models into a single multitask model that can perform well on both in-domain and out-of-domain scenarios. In this section, we evaluate the performance of TIES-MERGING with other methods across multiple different experimental settings.

**Merging PEFT Models.** Consider the setting where task vectors are computed based on parameters introduced during parameter-efficient fine-tuning. Specifically, we focus on $(IA)^3$ [43], a PEFT method that scales the base model activations with learned vectors. We follow Liu et al. [43] and use T0-3B [66] as the base model and finetune $(IA)^3$ models on the train split of eleven datasets including sentence completion (COPA [61], H-SWAG [88], and Story Cloze [68] datasets), natural language inference (ANLI [49], CB [44], and RTE [11]), coreference resolution (WSC [37] and Winogrande [64]), and word sense disambiguation (WiC [53]). When fine-tuning $(IA)^3$ parameters added to the T0-3B model, we use prompt templates from the Public Pool of Prompts (P3 [5]) to convert each example in each dataset to a prompted text-to-text format where each label corresponds to a different string. For experiments with $(IA)^3$, for each dataset, we report the median score across all templates.

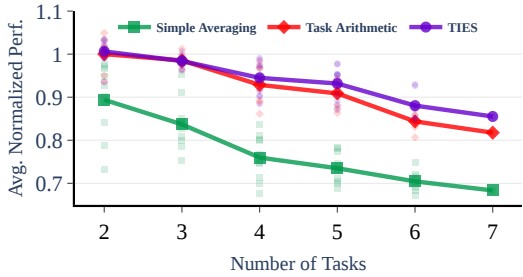

| Model | T5-Base | T5-Large |
|---|---|---|
| **Zeroshot** | 31.1 | 27.6 |
| **Simple Averaging** [9, 82] | 31.7 | 30.4 |
| **Fisher** [45] | 33.8 | 32.0 |
| **RegMean** [31] | 34.3 | 36.0 |
| **Task Arithmetic** [29] | 31.9 | 32.3 |
| **TIES-MERGING** | **35.3** [+1.0] | **40.4** [+4.4] |

Table 2: **TIES-MERGING generalizes better.** Out of Distribution Generalization for T5-Base and T5-Large on six held-out tasks.

Figure 5: **TIES-MERGING scales better.** Average performance when merging a different number of tasks.

Table 1 using TIES-MERGING to merge models trained with (IA)$^3$ exceeds the performance of all other merging methods – with a validation set, TIES-MERGING shows an average enhancement of 2.5% across 11 tasks compared to the top baseline. For detailed results, refer to Appendix Table 8.

**Merging Fully Finetuned Vision Models.** For image classification, we adhere to the experimental setting from Ilharco et al. [29, 28]. We employ two variants of the CLIP model [56] with ViT-B/32 and ViT-L/14 models [14] as visual encoders. Subsequently, we finetune the visual encoder on the eight tasks derived from Ilharco et al. [28, 29], Radford et al. [56] while keeping the text encoder fixed. This setting considers a variety of classification domains such as remote sensing, traffic classification, and satellite imagery recognition. Specifically, we work with the following datasets: Cars [35], DTD [10], EuroSAT [24], GTSRB [71], MNIST [36], RESISC45 [8], SUN397 [84], and SVHN [47].

Table 1 shows that using TIES-MERGING to merge fully fine-tuned ViT-B/32 and ViT-L/14 models leads to an average improvement of 1.8% and 1.5% over 8 tasks, given the availability of a validation set. In the absence of a validation set, TIES-MERGING improves by 6.6% and 2.7% over other methods for ViT-B/32 and ViT-L/14, respectively. Notably, TIES-MERGING without validation outperforms Task Arithmetic [29] with validation by 2.3% and 1.5% for ViT-B/32 and ViT-L/14. For more detailed results, refer to Appendix Table 11 and 12.

**Merging Fully Finetuned NLP Models.** For the NLP domain, we use the T5-base and T5-large [57] models, which are encoder-decoder transformers [77] pretrained via masked language modeling on a large text corpus. We finetune both T5-base and T5-large on seven tasks: question answering (QASC [33], WikiQA [87], and QuaRTz [73]), Paraphrase Identification (PAWS [90]), Sentence Completion (Story Cloze [68]), and Coreference Resolution (Winogrande [64] and WSC [37]).

Table 1 shows that using TIES-MERGING on T5-base and T5-large models with a validation set produces an improvement of 0.7% and 3.6% respectively over 7 tasks compared to the state-of-the-art. Moreover, for T5-large TIES-MERGING without validation outperforms all baselines (even with a validation set) by 1.1%. For more detailed results, refer to Appendix Table 9 and 10.

**Out-of-Domain Generalization.** In many use-cases, multitask models are used for their ability to generalize better to domain shift. Hence, we use the T5-base and T5-large models merged on the seven in-domain datasets from the previous experiments and evaluate them on six held-out datasets from T0 mixture [65] to measure out-of-domain generalization. Specifically, we report the average performance over the following tasks and datasets: Cosmos QA [27], Social IQA [67], and QuAIL [62] for question answering; WiC [53] for word sense disambiguation; and COPA [61], and H-SWAG [88] for sentence completion. Table 2 shows that TIES-MERGING outperforms the strongest baseline for both T5-base and T5-Large by 1.0% and 4.4% respectively, demonstrating better out-of-domain generalization. For more elaborate results please refer to Appendix B.6 and Table 13 and 14.

**Merging Different Number of Tasks.** We evaluate the performance of the merged model on the in-domain tasks as we vary the number of tasks being merged. In Fig. 5, we normalize the accuracy of each task by its fine-tuned model performance and report the average normalized accuracy on the in-domain tasks. We compare with the strongest baseline – Task Arithmetic [29] – as well as

| | RTE | MRPC | WNLI |
|---|---|---|---|
| **Averaging** | 59.9 | 78.2 | 56.3 |
| **Fisher** | 65.7 | 81.4 | 52.1 |
| **Ensembling** | 70.8 | 86.0 | 45.1 |
| **Task Arithmetic** | 71.8 | 86.0 | **59.2** |
| **TIES-MERGING** | **72.2** | **86.8** | 58.8 |

| **Init Method** | RTE | MRPC | WNLI |
|---|---|---|---|
| **PTM Init** | 66.4 | 81.8 | **56.3** |
| **Average** | 75.8 | 86.5 | **56.3** |
| **Task Arithmetic** | 78.3 | 86.2 | 50.7 |
| **TIES-MERGING** | **80.1** | **88.0** | 54.9 |

Table 3: **Model soups experimental setup. TIES improves performance when merging checkpoints on the same tasks.** For each task, we merge 10 checkpoints from Huggingface hub and evaluate on the one task they were trained on.

Table 4: **A TIES-merged model is a better initialization for finetuning.** For each task, we merge the checkpoints from the 7 other GLUE tasks and then finetune and evaluate on the selected task.

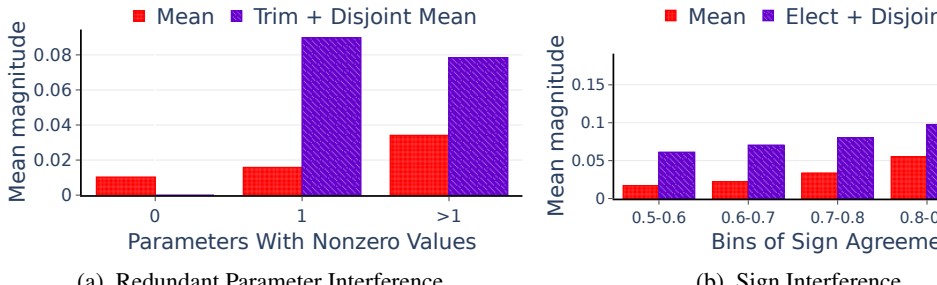

(a) Redundant Parameter Interference.

(b) Sign Interference.

Figure 6: **Trimming Parameters and Electing Signs prevents interference.** Demonstration of parameter interference between different models and its impact on parameter values. The Standard Mean (red) shrinks magnitudes and does it more when there is less agreement on the sign (right) or the parameter is influential for multiple tasks (left).

simple averaging [82]. Each data point signifies merging a subset of the tasks, with the solid line representing the mean performance across multiple subsets. For similar results with the T5-base model, please refer to Appendix C.5 and Figure 13.

From Fig. 5, we observe the following: (1) As the number of merged tasks increases, the performance of all methods decreases. (2) When merging two tasks, both TIES-MERGING and Task Arithmetic achieve an average normalized accuracy close to one, indicating negligible performance loss. In contrast, Simple Averaging suffers from a 10% performance drop. (3) As the number of tasks increases, the merging performance of Task Arithmetic declines more rapidly than TIES-MERGING. This suggests that task interference is present when merging multiple tasks and that TIES-MERGING is more effective at mitigating this issue.

**Merging Checkpoints of the Same Task For Better Robustness**  We perform additional experiments to merge multiple checkpoints trained on the same task (as done in ModelSoups [82]) to see if it can improve robustness. Typically, ensembling is used to combine different models on the same task for better generalization. We use the experimental setting and the code from Fisher Merging [45] to merge top-10 fine-tuned base sized BERT models from huggingface for RTE, MRPC, and WNLI datasets from GLUE. From the results presented in Table 3, we observe that TIES-MERGING works the best in all cases except WNLI, where it only slightly underperforms Task Vectors. Notably, TIES-MERGING provides a dramatic boost over both Fisher Merging, averaging, and outperforms *ensembling* in all cases. Moreover, in Appendix B.4, we show that interference exists even between differently finetuned checkpoints of the same tasks.

**Merging Models for Better Initialization.**  Next, we perform experiments following the setting [9], where we merge checkpoints from different tasks for a better initialization when fine-tuning on a downstream task. We take the finetuned `bert-base-uncased` checkpoints for 8 GLUE [78] tasks (wnli, sst2, rte, qnli, mrpc, cola, mnli, qqp) from Huggingface [81]. We consider three of these GLUE tasks (RTE, MRPC, WNLI) as our downstream tasks. When fine-tuning on a particular

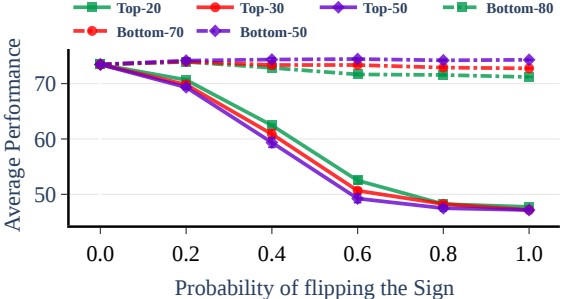

| Method | Average |
|---|---|
| **Fine-Tuned** | 71.4 |
| **Multitask** | 73.1 |
| **Averaging** [9, 82] | 58.0 |
| **Task Vectors** [29] | 63.9 |
| **TIES-MERGING** | **66.4** |
| **TIES-MERGING (Oracle Sign)** | **72.0** [+5.6] |

Figure 7: **Flipping the signs of high magnitude parameters leads to catastrophic performance drops.** Average Performance when flipping the directions of Top-$k\%$ and Bottom-$k\%$ parameters for each task. We report the results averaged over eleven $(IA)^3$ tasks.

Table 5: **TIES-MERGING can perform close to multitask models if the signs can be estimated correctly.** We use the signs from the multitask vector as the elected sign and perform merging and report the performance.

downstream task (say RTE), we merge all the checkpoints from the other seven tasks together (apart from the chosen task). From Table 4, we find that TIES-MERGING works well in this setting and outperforms all other merging methods by a significant margin (apart from Averaging for WNLI).

# 7 Additional Results and Analysis

## 7.1 Types of Interference and Their Effect on Merging

**(a) Importance of Removing Redundant Parameters.** To better disentangle the effect of redundant parameters on the resulting magnitude of merged parameters, we separate the parameters into three groups: redundant parameters (using a trimming threshold of $20\%$), parameters that are influential to exactly one model, and parameters that are influential to more than one model. We then compare the parameter values when they are directly merged versus when they are first trimmed and then (disjointly) merged without electing signs. Specifically, we only take the mean of non-trimmed values. The results are shown for the PEFT setting in Fig. 6a, which demonstrates that redundant parameters cause interference. Specifically, we find that when a parameter is not an influential parameter for any of the task-specific models, the mean value is low, and therefore may be considered noise. However, when only one model sees the parameter as influential, the merged value can still be low since other models assign a small value to this parameter. The merged value is larger when more models see the parameter as influential. When trimming, we see this interference is mostly avoided, and the average size is mostly the same whether one or more models consider a parameter influential. This is because we remove the effect of noisy parameters that unnecessarily decrease the magnitude (see ○ in Fig. 2). In the Appendix B.5, we bring a more detailed view, including a comparison to applying TIES-MERGING and show reducing interference encourages diversity in specific parameter values (std) together with the similarity of their influence (mean).

**(b) Importance of Resolving Sign Interference.** To quantify the impact of sign interference, we group the parameters by their *sign agreement*. A value of $0.5$ indicates an equal number of positive and negative signs for a given parameter across different models, whereas 1 implies all the parameters have the same sign. We compare the parameter values when those are merged, or when sign disagreement is first resolved by election and then they are (disjointly) merged. The results in the PEFT setting are shown in Fig. 6b, where we demonstrate that the ELECT step preserves the relative parameter magnitudes to avoid sign interference. Specifically, we find that resolving signs increases the overall parameter magnitudes across different ranges of sign agreements. Parameters with low agreement tend to be smaller on average regardless of the interference. One potential cause could be that the sign from noisy parameters pulls the average down, as seen in Fig. 6a. We show in Appendix B.5 that combining both methods indeed reduces some of the difference, but not all, suggesting that a high agreement is correlated with overall influential parameters.

## 7.2 Relevance of Signs of the Top-$k\%$ Parameters

In this experiment, we work with the (IA)$^3$ models and aim to quantify the importance of the top-$k\%$ parameters and their directions on a task's performance. For each task vector $\tau_t$, we flip the direction of each of the top-$k\%$ parameters (by magnitude) with a probability $p$ to obtain $\tilde{\tau}_t$. Flipping the direction is done by multiplying the parameters with $-1$. Then we add back this direction flipped $\tilde{\tau}_t$ to the initialization to get $\tilde{\theta}_t = \theta_{\text{init}} + \tilde{\tau}_t$. Finally, we evaluate $\tilde{\theta}_t$ and compute the average performance over all tasks $t$ for each value of $k$ and $p$. As a baseline, we also flip the directions of the bottom $(100 - k)\%$ of the parameters. We report the results averaged over three independent runs.

In Fig. 7, we plot the average performance as a function of $p$, the probability of flipping the direction. A probability of $0$ means that the direction of none of the top-$k\%$ parameters is flipped and a value of $1$ means that the direction of all the top-$k\%$ parameters is flipped. For the top-$20/30\%$ of the parameters (solid lines), we observe that the performance monotonically decreases as we increase the probability of flipping the direction. In contrast, flipping the directions of the bottom-$80/70\%$ of the parameters (dashed lines) has little impact on the performance. These results establish the importance of having the right directions for the parameters with a high magnitude and show the catastrophic performance drop that happens with incorrect directions.

## 7.3 Ablation of TIES-MERGING Components

We perform ablations on the individual components of TIES-MERGING to assess their importance. In Table 6, we start with TIES-MERGING and remove one component at a time and report the performance on the validation set for full model merging (T5-base) and merging PEFT models ((IA)$^3$ on T03B). Removing elect while keeping the disjoint mean refers to taking the mean of values with signs $+1$ and $-1$ but not including the $0$ values of the trimmed task vectors in the mean. Removing disjoint mean but trimming and electing refers to taking the mean

| Method | T5-base | (IA)$^3$ |
|---|---|---|
| **TIES-MERGING** | **74.5** | **70.7** |
| − TRIM | 73.0 | 70.6 |
| − ELECT | 73.1 | 69.6 |
| − DISJOINT MEAN | 72.6 | 67.5 |
| − SCALE | 72.0 | 65.5 |

Table 6: Ablation on all the steps of TIES-MERGING.

of the values with the elected signs and the $0$ for the trimmed values. Removing scaling means using $\lambda = 1$. Table 6 shows that all components of the method are crucial for optimal performance. Specifically, scaling and the disjoint mean emerge as the most crucial, causing performance declines of $2.5\%$ and $1.9\%$ in T5-base, and $5.2\%$ and $3.2\%$ in (IA)$^3$, respectively.

## 7.4 Importance of Estimating Correct Signs When Merging Models

Given the importance of sign vectors, we now aim to understand the performance that can be obtained by TIES-MERGING if we can use the oracle sign vector from the multitask model. To test this, we train a multitask (IA)$^3$ model, $\theta_{\text{mult}}$, on the eleven tasks under consideration (as in § 6). We then create the multitask vector $\tau_{\text{mult}}$ and the multitask sign vector $\gamma_{\text{mult}}$. Next, while merging models using TIES-MERGING, we assume access to the oracle multitask-sign-vector $\gamma_{\text{mult}}$. Hence, we skip the conflict resolution step and directly set $\gamma_m = \gamma_{\text{mult}}$. Surprisingly, from Table 5, we observe that when merging tasks by using the oracle sign vector, we get a performance of $72\%$ compared to $73.1\%$ for the multitask trained model. Moreover, on average the merged model performs better task-specific models. This implies that if we can obtain the correction directions for the merged model, then we can closely bridge the gap to multitask models. In Appendix B.1 and Table 7, we attempt to estimate the multitask-sign-vector by using limited data.

# 8 Conclusion

We introduced TIES-MERGING to address interference when merging models. TIES-MERGING trims low-magnitude changes in fine-tuned model's values and then resolves sign disagreements across the models being merged. We found experimentally that TIES-MERGING enhances the performance of the merged multitask model across various settings and domains, despite being simple with fixed hyperparameters. Our study also sheds light on the impact of different types of interference on model parameters and emphasizes the importance of signs in the merging process. For some discussion on limitations and future directions please refer to Appendix A.

## Acknowledgements

We thank Yi-Lin Sung, Shiyue Zhang, Archiki Prasad, and the reviewers for their valuable feedback on this paper. This work is supported by NSF-AI Engage Institute DRL211263, NSF-CAREER Award 1846185, DARPA MCS Grant N66001-19-2-4031, and NSF Grant 2145822. The views, opinions, and/or findings contained in this article are those of the authors and not of the funding agency.

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

# Appendix for TIES-MERGING

## A  Limitations and Future Works

Our works share the same general limitations of existing merging methods, like (1) a limited theoretical understanding of why and when weight interpolation works, what are the important underlying factors, and its proper connections with mode connectivity. Recent works like [50] have demonstrated interesting relationships between weight disentanglement and mergingability of models; (2) that merging relies on common initialization and model architecture; and (3) merging individual-task models to create a multitask still lags behind the simultaneous multitask training. Moreover, it is not clear how to select the checkpoints for merging in order to create multitask models useful for specific domains. In addition, while our method provides a way to choose signs when merging task vectors, we still find that using the signs from a multitask model performs better. Some potential future works include figuring out a good way to estimate multitask signs without having access to the multitask model as this has the potential to bridge the gap between multitask merging and multitask training (as demonstrated in Section 7.4).

## B  Additional Results

| Method | Estimating Sign | | | Average |
|---|---|---|---|---|
| | **Multitask** | **Samples** | **Init.** | |
| **Fine-Tuned** | - | - | - | 71.4 |
| **Multitask** | - | - | - | 73.1 |
| **Averaging** [9, 82] | - | - | - | 58.0 |
| **Task Vectors** [29] | - | - | - | 63.9 |
| **TIES-MERGING** | - | - | - | **66.4** |
| | ✓ | 32 | scratch | 66.5 [+0.1] |
| **TIES-MERGING** | ✓ | 32 | mean | 67.7 [+1.2] |
| | ✓ | All | scratch | **72.0** [+5.6] |

Table 7: **Merging Performance can be improved by estimating the Sign Vector by performing few-shot multitask training.** We use the estimated sign as the elected sign and perform merging.

### B.1  Enhancing Performance by Estimating the Multitask Sign Vector.

Considering the findings, we inquire whether it is possible to efficiently acquire multitask sign vectors without extensive multitask training. Our proposal involves utilizing a limited number of validation samples from each task to cheaply train a multitask model and subsequently derive the relevant sign vector. We create two multitask $(IA)^3$ models: one developed from scratch and another initialized using the average of task-specific $(IA)^3$ models intended for merging. We use 32 validation examples from each task to train this model. In Table 5, we notice using the sign vector from the fewshot multitask model initialized with mean yielded a performance increase of $3.8\%$ and $1.3\%$ compared to Task Arithmetic and TIES-MERGING. Interestingly, training fewshot multitask training from scratch did not yield significant improvements over TIES-MERGING without sign estimation. We believe that exploring this area further may result in improved merging techniques.

### B.2  Effect of Hyper-Parameters $\lambda$ and K on the Performance.

In Figure 8 (left and middle), we plot the effect of $\lambda$ on the performance when merging T5-base and T5-large models trained on GLUE (Similar to Table-1). For TIES-MERGING, we vary around the value 1 because TIES takes the mean of task vectors, whereas task arithmetic adds up the task vectors. Hence, a value of 1 for TIES is similar to using $\frac{1}{\#tasks}$ for Task Arithmetic [29]. The range of 0.8-1.8 for TIES was selected based on preliminary experiments on the PEFT setting (as mentioned in Section 5). We find that TIES-MERGING is much less sensitive to changes in (with an accuracy range of 68-75% across the considered values of $\lambda$) compared to Task Arithmetic (with an accuracy

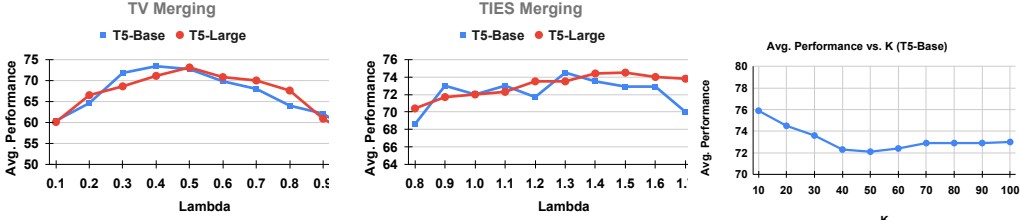

Figure 8: **Performance as a function of hyperparameters**. For more details please refer to the response to our **general response.**

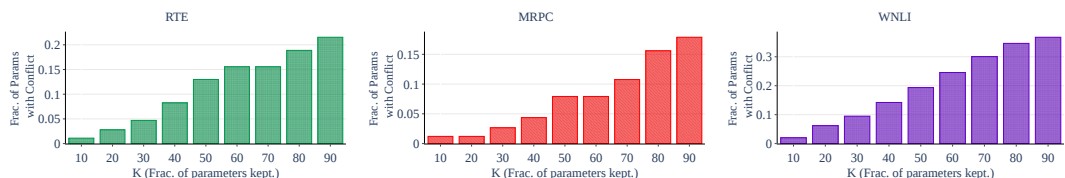

Figure 9: **Sign conflict increases as we trim less parameters.** For each task, we merge 10 different checkpoints from hunggingface hub and plot the sign conflict as a function of keeping only the top-k% parameters.

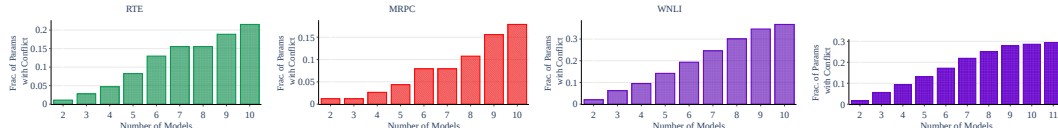

Figure 10: **Sign Conflict exists even when merging multiple checkpoints for the same task.** The first three plots are for RTE, MRPC, WNLI datasets when merging 10 Huggingface checkpoints, while the last one is when merging different tasks (Figure 4 from the main paper).

range of 55-75). Figure 8 (right) demonstrates the effect of $k$ as we increment the value of $k$ in steps of 10 and skip $k = 0$ as that would select no parameters. We observe that as $k$ increases the performance drops and then saturates. However, we would like to note that this curve might change based on the distribution of the parameter values in the task vector.

### B.3 Sign Conflict Increases as We Trim Less Parameters

In Figure 9, we merge 10 `bert-base-uncased` checkpoints from huggingface finetuned on for three different glue tasks (RTE, MRPC, and WNLI) and plot the sign conflict as a function of $k$. As we keep more and more parameters, the sign conflict increases and reaches almost 80%. This is also expected as there are many more nonzero parameters that can create conflict even if their magnitude is small.

### B.4 Sign Conflicts Exists Between Different Checkpoints for the Same Task

In Figure 10, we show that sign conflicts exist even within models trained on the same task. We plotted the sign conflict (similar to Figure 4) between the 10 checkpoints of RTE, MRPC, and WNLI from Huggingface. As the number of checkpoints increases, sign conflict increases. We also compare this with the sign interference when merging different task checkpoints and find a similar degree of interference in all of these cases. Hence, sign conflicts exist even within models trained on the same dataset. We suspect that this is because models are highly overparameterized and hence there are multiple subnetworks (subsets of parameters) that could lead to the same performance where different finetuning runs update the same parameters in different directions.

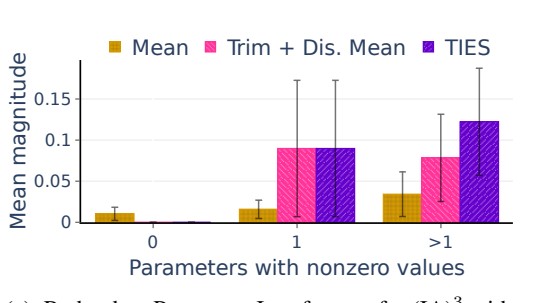

(a) Redundant Parameter Interference for (IA)$^3$ with STD.

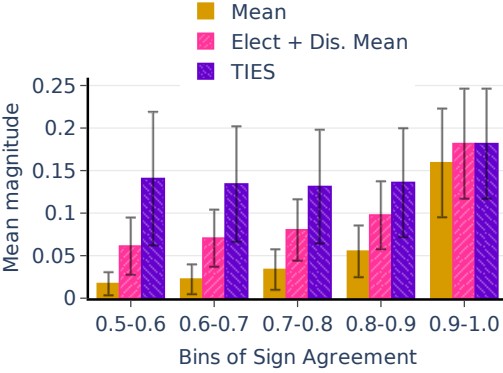

(b) Sign Interference for (IA)$^3$ with STD.

Figure 11: Effect of different types of Merging on the Magnitudes of the Parameters. Here we additionally compare with TIES-MERGING and also provide the standard deviation of parameter values. A high std implies that there is some diversity in magnitude values across different parameters.

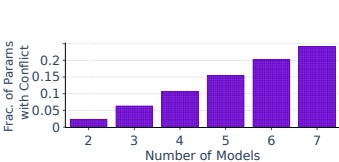

(a) Fraction of Parameters with Sign conflicts for T5-Base model versus number of models.

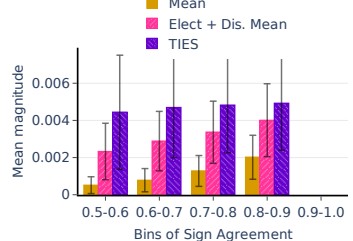

(b) Redundant Parameter Interference for T5-Base with STD.

(c) Sign Interference for T5-Base model with STD.

Figure 12: Plots for T5-Base model.

| Method | Validation | Average | rte | cb | winogrande | wic | wsc | copa | h-swag | story cloze | anli-r1 | anli-r2 | anli-r3 |
|---|---|---|---|---|---|---|---|---|---|---|---|---|---|
| **Zeroshot** | - | 55.3 | 79.8 | 46.4 | 52.8 | 54.1 | 45.2 | 85 | 36.1 | 91 | 39.7 | 37.6 | 40.5 |
| **Fine-Tuned** | - | 71.4 | 82.7 | 95.8 | 75.1 | 71.7 | 65.3 | 85.3 | 44.4 | 94.9 | 70.2 | 46.5 | 53 |
| **Multitask (All, scratch)** | - | 73.1 | 88.6 | 95.8 | 75.5 | 61.1 | 80.6 | 94.1 | 42.3 | 97.6 | 70.5 | 49.8 | 47.7 |
| **Multitask (32, scratch)** | - | 60.9 | 74.9 | 79.2 | 59.3 | 49.2 | 63.9 | 80.9 | 39.5 | 91.6 | 49.4 | 41.9 | 40.1 |
| **Multitask (32, mean)** | - | 65.2 | 79.8 | 83.3 | 61.6 | 54.2 | 66.7 | 85.3 | 41.1 | 94.4 | 58.1 | 46.0 | 46.5 |
| **Averaging** | ✗ | 58 | 81.2 | 58.3 | 53.8 | 55.2 | 53.5 | 80.9 | 40.1 | 92.5 | 43.3 | 39.2 | 40.2 |
| **Task Arithmetic** | ✗ | 59.2 | 76.5 | 79.2 | 57.7 | 51.6 | 51.4 | 66.2 | 31.4 | 91.6 | 59.8 | 47.5 | 48.2 |
| **TIES-MERGING** | ✗ | 64.9 | 81.2 | 87.5 | 60.8 | 59.9 | 58.3 | 80.2 | 42.6 | 91.1 | 58.1 | 46.5 | 47.4 |
| **Fisher Merging** | ✓ | 62.2 | 83.3 | 83.3 | 56.7 | 54.2 | 58.3 | 83.1 | 42.2 | 94.1 | 45.9 | 41.0 | 42.2 |
| **RegMean** | ✓ | 58 | 81.2 | 58.3 | 53.8 | 55.2 | 53.5 | 80.9 | 40.1 | 92.5 | 43.3 | 39.2 | 40.2 |
| **Task Arithmetic** | ✓ | 63.9 | 74.1 | 83.3 | 62.8 | 49.1 | 49.3 | 87.5 | 41.5 | 95.3 | 60.8 | 49.4 | 50.0 |
| **TIES-MERGING** | ✓ | 66.4 | 78.0 | 83.3 | 67.9 | 57.6 | 59.7 | 81.7 | 42.8 | 90.3 | 66.9 | 51.3 | 51.1 |

Table 8: Test set performance when merging IA3 models on eleven tasks. Please refer to Section 6 for experimental details.

## B.5 Detailed Results for Types of Interference and Their Effect on Merging

In Section 7.1 and Figure 6, we showed the effect of redundant parameters and sign conflicts on parameter magnitudes when comparing simple averaging vs disjoint mean after either trimming or electing and showed that performing these operations helps with the parameter magnitudes. In Figure 11, we additionally compare with TIES-MERGING and show that performing both trimming and electing usually results in higher magnitude and also higher standard deviation in parameter magnitudes. Higher std denotes that all parameter values in the merged model are the same and that there is a significant variation in the magnitude which is in contrast to simple averaging as it decreases the magnitude of not redundant parameters and reduces the magnitude of the influential parameters in the merged model. Similar plots for the T5-base model are provided in Figure 12.

## B.6 Comprehensive Task-Level Results

We provide the task level for all the in-domain evaluation experiments in the main Table 1. Table 8, 9, 10, 11, and 12 provide the task level results IA3 [43], T5-Base, T5-Large [58], ViT-B/32, and ViT-L/14 [14] respectively. The task level results of the out-of-domain experiments for T5-Base and T5-Large can be found in Table 13, and 14. Lastly, Figure 13, shows the scaling of the T5-Base model as we merge different numbers of tasks.

| Method | Vailidation | Average | paws | qasc | quartz | story_cloze | wiki_qa | winogrande | wsc |
|---|---|---|---|---|---|---|---|---|---|
| Zeroshot | - | 53.5 | 49.9 | 35.8 | 53.3 | 48.1 | 76.2 | 50 | 61.1 |
| Fine-tuned | - | 82.8 | 94.3 | 98.3 | 80.4 | 84.7 | 95.5 | 64.1 | 62.5 |
| Multitask | - | 83.6 | 94 | 97.9 | 82.5 | 86.7 | 95 | 64.1 | 65.3 |
| Averaging | ✗ | 65.9 | 66.4 | 82.6 | 60.2 | 49.5 | 94.1 | 50.4 | 58.3 |
| Task Arithmetic | ✗ | 73.9 | 73.3 | 93.5 | 68.2 | 76.5 | 93.7 | 55.5 | 56.9 |
| TIES-MERGING | ✗ | 69.7 | 74 | 83.3 | 70.3 | 64.2 | 84.7 | 55.9 | 55.6 |
| Fisher Merging | ✓ | 68.9 | 69.3 | 85.7 | 63.6 | 56.4 | 93.8 | 50.9 | 62.5 |
| RegMean | ✓ | 71.2 | 76.8 | 96.2 | 62.5 | 55 | 94.8 | 51.9 | 61.1 |
| Task Arithmetic | ✓ | 73.2 | 73.4 | 93.3 | 67.1 | 71.7 | 94.1 | 52.9 | 59.7 |
| TIES-MERGING | ✓ | 73.9 | 79.3 | 88.6 | 71.8 | 72.9 | 82.5 | 61.3 | 61.1 |

Table 9: Test set performance when merging T5-base models on seven tasks. Please refer to Section 6 for experimental details.

| Method | Validation | Average | paws | qasc | quartz | story_cloze | wiki_qa | winogrande | wsc |
|---|---|---|---|---|---|---|---|---|---|
| Zeroshot | - | 51.7 | 55.4 | 14.3 | 54.1 | 54.1 | 71 | 49.3 | 63.9 |
| Fine-tuned | - | 88.8 | 94.4 | 98.9 | 87.8 | 90.8 | 96 | 74.7 | 79.2 |
| Multitask | - | 88.1 | 94.2 | 98.5 | 89.3 | 92 | 95.4 | 73.5 | 73.6 |
| Averaging | ✗ | 59.6 | 61.3 | 82.6 | 70.5 | 53.7 | 63.2 | 49.7 | 36.1 |
| Task Arithmetic | ✗ | 73.5 | 79.2 | 96.8 | 80.2 | 83.6 | 58.6 | 60.2 | 55.6 |
| TIES-MERGING | ✗ | 74.4 | 80.5 | 96.2 | 81.8 | 78.6 | 62 | 61.9 | 59.7 |
| Fisher Merging | ✓ | 64.6 | 60.4 | 81.7 | 75 | 60.1 | 88.6 | 50 | 36.1 |
| RegMean | ✓ | 73.2 | 86 | 96.9 | 80.7 | 78.6 | 82.6 | 51.8 | 36.1 |
| Task Arithmetic | ✓ | 73.3 | 77.8 | 96 | 78.6 | 86.4 | 59.1 | 62.3 | 52.8 |
| TIES-MERGING | ✓ | 76.9 | 81.5 | 96.2 | 80.1 | 83.6 | 64.9 | 66.5 | 65.3 |

Table 10: Test set performance when merging T5-large models on seven tasks. Please refer to Section 6 for experimental details.

| Method | Validation | Average | SUN397 | Cars | RESISC45 | EuroSAT | SVHN | GTSRB | MNIST | DTD |
|---|---|---|---|---|---|---|---|---|---|---|
| Individual | - | 90.5 | 75.3 | 77.7 | 96.1 | 99.7 | 97.5 | 98.7 | 99.7 | 79.4 |
| Multitask | - | 88.9 | 74.4 | 77.9 | 98.2 | 98.9 | 99.5 | 93.9 | 72.9 | 95.8 |
| Averaging | ✗ | 65.8 | 65.3 | 63.4 | 71.4 | 71.7 | 64.2 | 52.8 | 87.5 | 50.1 |
| Task Arithmetic | ✗ | 60.4 | 36.7 | 41 | 53.8 | 64.4 | 80.6 | 66 | 98.1 | 42.5 |
| TIES-MERGING | ✗ | 72.4 | 59.8 | 58.6 | 70.7 | 79.7 | 86.2 | 72.1 | 98.3 | 54.2 |
| Fisher Merging | ✓ | 68.3 | 68.6 | 69.2 | 70.7 | 66.4 | 72.9 | 51.1 | 87.9 | 59.9 |
| RegMean | ✓ | 71.8 | 65.3 | 63.5 | 75.6 | 78.6 | 78.1 | 67.4 | 93.7 | 52 |
| Task Arithmetic | ✓ | 70.1 | 63.8 | 62.1 | 72 | 77.6 | 74.4 | 65.1 | 94 | 52.2 |
| TIES-MERGING | ✓ | 73.6 | 64.8 | 62.9 | 74.3 | 78.9 | 83.1 | 71.4 | 97.6 | 56.2 |

Table 11: Test set performance when merging ViT-B/32 models on eight tasks. Please refer to Section 6 for experimental details.

# C Implementation Details

## C.1 Compute Resources Used and Runtimes

We executed all our experiments on Nvidia A6000 GPUs equipped with 48GB RAM. Single-task $(IA)^3$ models for eleven tasks required 1-2 hours per model, while the multitask vector took around 24 hours on four GPUs. The T5-Base and T5-Large models, based on dataset size, needed between 15 minutes and 2 hours per task, and approximately eight hours for the multitask checkpoint. Vision

| Method | Validation | Average | SUN397 | Cars | RESISC45 | EuroSAT | SVHN | GTSRB | MNIST | DTD |
|---|---|---|---|---|---|---|---|---|---|---|
| Fine-tuned | - | 94.2 | 82.3 | 92.4 | 97.4 | 100 | 98.1 | 99.2 | 99.7 | 84.1 |
| Multitask | - | 93.5 | 90.6 | 84.4 | 99.2 | 99.1 | 99.6 | 96.3 | 80.8 | 97.6 |
| Averaging | ✗ | 79.6 | 72.1 | 81.6 | 82.6 | 91.9 | 78.2 | 70.7 | 97.1 | 62.8 |
| Task Arithmetic | ✗ | 83.3 | 72.5 | 79.2 | 84.5 | 90.6 | 89.2 | 86.5 | 99.1 | 64.3 |
| TIES-MERGING | ✗ | 86 | 76.5 | 85 | 89.3 | 95.7 | 90.3 | 83.3 | 99 | 68.8 |
| Fisher Merging | ✓ | 82.2 | 69.2 | 88.6 | 87.5 | 93.5 | 80.6 | 74.8 | 93.3 | 70 |
| RegMean | ✓ | 83.7 | 73.3 | 81.8 | 86.1 | 97 | 88 | 84.2 | 98.5 | 60.8 |
| Task Arithmetic | ✓ | 84.5 | 74.1 | 82.1 | 86.7 | 93.8 | 87.9 | 86.8 | 98.9 | 65.6 |
| TIES-MERGING | ✓ | 86 | 76.5 | 85 | 89.4 | 95.9 | 90.3 | 83.3 | 99 | 68.8 |

Table 12: Test set performance when merging ViT-L/14 models on eight tasks. Please refer to Section 6 for experimental details.

models ViT-B/32 and ViT-L/14 were utilized, as supplied by Ilharco et al. [29].[2] Merge experiments were efficient, with evaluations consuming less than 2 minutes for the T5-Base, T5-Large, ViT-B/32, and ViT-L/14 experiments. The assessment of $(IA)^3$ models, due to the necessity of using multiple templates from prompt sources and median result calculations across all templates, required approximately one hour per 11 dataset evaluation.

| Model | Average | cosmos_qa | social_iqa | quail | wic | copa | h-swag |
|---|---|---|---|---|---|---|---|
| PAWS | 35.9 | 18.8 | 25 | 24.8 | 68.8 | 56.2 | 21.9 |
| QASC | 34.9 | 15.6 | 21.9 | 25.1 | 75 | 53.1 | 18.8 |
| QUARTZ | 37.4 | 31.2 | 18.8 | 24.3 | 71.9 | 59.4 | 18.8 |
| Story Cloze | 35 | 6.2 | 25 | 25.6 | 75 | 65.6 | 12.5 |
| Wiki QA | 24.5 | 18.8 | 21.9 | 24.9 | 28.1 | 43.8 | 9.4 |
| Winogrande | 28.3 | 18.8 | 25 | 25.7 | 34.4 | 43.8 | 21.9 |
| WSC | 31.7 | 21.9 | 21.9 | 24.6 | 62.5 | 46.9 | 12.5 |
| Pretrained | 31.1 | 21.9 | 18.8 | 24.1 | 65.6 | 43.8 | 12.5 |
| Averaging | 31.7 | 21.9 | 21.9 | 24.6 | 68.8 | 37.5 | 15.6 |
| Fisher Merging | 33.8 | 15.6 | 21.9 | 24.9 | 65.6 | 53.1 | 21.9 |
| Task Arithmetic | 31.9 | 15.6 | 31.2 | 25.7 | 28.1 | 68.8 | 21.9 |
| RegMean | 34.3 | 23.1 | 28.1 | 24.9 | 48.4 | 62.5 | 18.8 |
| TIES-MERGING | 35.3 | 21.9 | 25 | 25.7 | 50 | 65.6 | 23.8 |

Table 13: Out-of-Distributon performance of T5-Base model checkpoints on six tasks. Please refer to Section 6 for experimental details.

## C.2 Employed Datasets and Associated Licences

We use the following datasets in the paper with the following licenses. ANLI [49], WiC [53], WSC [37], and Story Cloze [68], QuaRTz [73], Cars [35], GTSRB [71] are under Creative Commons License. Winogrande [64], QASC [33] are under Apache license. COPA [61] is under a BSD-2 Clause license. H-SWAG [88], EuroSAT [24], is under MIT Licence. MNIST [36] is under Gnu General Public License. We could not find the licences of DTD [10], RESISC45 [8], SUN397 [84], SVHN [47], CB [44], RTE [11]), and PAWS [90] but they are publically for research use.

## C.3 Details of the Motivation Experiments

For both Figure 3, and 4 in Section 3, we perform experiment on the eleven $(IA)^3$ models used in our PEFT merging experiments (§ 6). For a Figure similar to Fig. 4 demonstrating the fraction of parameters with a sign conflict for T5-base model, please refer to Fig. 12a.

---

[2]https://github.com/mlfoundations/task_vectors#checkpoints

| Model | Average | cosmos_qa | social_iqa | quail | wic | copa | h-swag |
|---|---|---|---|---|---|---|---|
| **PAWS** | 32.3 | 25 | 28.1 | 25.6 | 56.2 | 46.9 | 12.5 |
| **QASC** | 33.4 | 21.9 | 28.1 | 25.5 | 43.8 | 62.5 | 18.8 |
| **QUARTZ** | 28.7 | 25 | 25 | 25.1 | 25 | 53.1 | 18.8 |
| **Story Cloze** | 32.1 | 21.9 | 34.4 | 26.8 | 46.9 | 53.1 | 9.4 |
| **Wiki QA** | 27.1 | 25 | 28.1 | 25.2 | 28.1 | 46.9 | 9.4 |
| **Winogrande** | 32.4 | 31.2 | 18.8 | 25.6 | 43.8 | 62.5 | 12.5 |
| **WSC** | 29.7 | 25 | 25 | 25.1 | 37.5 | 56.2 | 9.4 |
| **Pretrained** | 27.6 | 21.9 | 21.9 | 24.9 | 28.1 | 56.2 | 12.5 |
| **Averaging** | 30.4 | 31.2 | 25 | 26.3 | 31.2 | 59.4 | 9.4 |
| **Fisher Merging** | 32 | 34.4 | 25 | 26.1 | 40.6 | 56.2 | 9.4 |
| **Task Arithmetic** | 33.3 | 21.9 | 34.4 | 24.6 | 40.6 | 59.4 | 18.8 |
| **RegMean** | 36 | 34.4 | 28.1 | 25.3 | 62.5 | 50 | 15.6 |
| **TIES-MERGING** | 40.4 | 31.2 | 43.8 | 26.6 | 59.4 | 59.4 | 21.9 |

Table 14: Out-of-Distributon performance of T5-Large model checkpoints on six tasks. Please refer to Section 6 for experimental details.

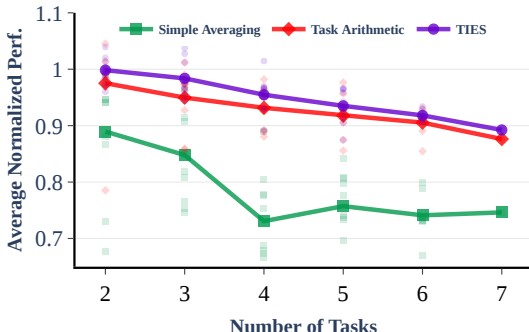

Figure 13: T5-Base with increasing number of task being merged. Average performance when merging a different number of tasks.

## C.4 Merging in the absence of the Validation Set

In our investigation into scenarios where a validation set is not available, we have devised a recipe and identified the optimal hyperparameters, employing the PEFT experimental procedure detailed in Section 6. This approach was applied to the eleven task-specific models presented in the same section, utilizing the TIES-MERGING method for tuning the hyperparameters. Our preliminary estimates for the hyperparameters were $k = 20$ and $\lambda$ close to 1. The hyperparameter search was conducted using the eleven task-specific (IA)$^3$ models, with $k \in \{10, 20, 30\}$, and $\lambda$ spanning from 0.8 to 3.0, in increments of 0.1. The results of this comprehensive search indicated an optimal value of $k = 20$, with values of $\lambda = 0.9$, $\lambda = 1.0$, and $\lambda = 1.1$ demonstrating equivalent performance. To maintain simplicity in our model, we chose a $\lambda$ value of 1. Thus, the final selection of parameters for TIES-MERGING is $k = 20$, signs based on mass, the disjoint mean, and a $\lambda$ value of 1.

## C.5 Merging Different Number of Tasks

Here we provide some additional details on the experiments when merging different numbers of tasks. In Fig. 5, we perform the experiment with T5-Large when merging the seven tasks considered in Tab. 1 and described in Sec. 6. The x-axis shows the different number of tasks being merged. Note that when merging $T$ tasks, we have a total of $\binom{7}{T}$ combinations. However, in our experiment, we sample at most 10 distinct combinations for each value of $T$. A similar plot for the T5-Base model is shown in Fig. 13.

In Figure 5, for each number of tasks we take at most 10 random subsets of the 8 tasks we were considering. The solid line is the average of the merged model's performance from these different runs. Below we provide the optimal $\lambda$ values for the different subsets of tasks we merged for both TIES-MERGING and Task Arithmetic, note that for averaging the $\lambda = \frac{1}{\#tasks}$ always. Each entry in the list is the optimal $\lambda$ for a particular subset of tasks selected on the validation set.

(2 tasks) TIES → [1.7, 1.9, 2, 2, 1.1, 1.5, 1.6, 1.8, 1.9, 1., 5]
(2 tasks) Task Arithmetic → [1, 0.9, 1, 1, 0.9, 1, 0.9, 0.9, 0.9, 1]

(3 tasks) TIES → [1.2, 2, 1.5, 1.9, 1.8, 1.7, 1.4, 2, 3, 1.9]
(3 tasks) Task Arithmetic → [1, 0.7, 0.7, 1, 1, 0.9, 0.7, 0.7, 0.9, 1]

(4 tasks) TIES → [1.5, 1.3, 1.3, 1.8, 2.3, 1.7, 1.8, 1.7, 1.9, 1.5]
(4 tasks) Task Arithmetic → [0.8, 0.7, 0.7, 0.7, 0.6, 0.7, 0.7, 0.8, 0.6, 0.7]

(5 tasks) TIES → [2, 2, 2, 1.8, 1.7, 2, 1.6, 2.1, 1.6, 1.3]
(5 tasks)Task Arithmetic → [0.7, 0.8, 0.6, 0.8, 0.7, 0.6, 0.6, 0.6, 0.6, 0.7]

(6 tasks) TIES → [1.6, 1.7, 1.7, 1.2, 1.7, 1.7, 1.5]
(6 tasks) Task Arithmetic → [0.6, 0.5, 0.5, 0.5, 0.7, 0.5, 0.6]

(7 tasks) TIES → [1.7]
(7 tasks) Task Arithmetic → [0.5]

## C.6 Training Details

In our research, we utilized two variants of the T5 model, specifically the T5-base and T5-large models, which were trained to a maximum of 75,000 steps. An effective training batch size of 1024 was implemented, alongside a learning rate (lr) of 0.0001. We instituted an early stopping mechanism with a patience threshold of 5 to prevent overfitting. During the training process, bfloat16 was adopted to curtail GPU memory expenditure, and the maximum sequence length was set at 128. In contrast, for the PEFT configuration of the (IA)$^3$ approach on the T0-3B model, we modified our parameters. An effective training batch size of 16 was deployed along with an evaluation batch size of 32, while maintaining the learning rate at 0.0001. To accommodate the model's complexity, the early stopping patience was augmented to 10. We do not use any lr scheduler and weight decay for any of our model training.

For the purpose of evaluation, we perform *rank classification*. In this method, the model's log probabilities for all potential label strings are ranked. The model's prediction is deemed accurate if the choice ranked highest aligns with the correct answer. It should be noted that rank classification evaluation can accommodate both classification tasks and multiple-choice tasks.

