# OpenReview forum: "TIES-Merging: Resolving Interference When Merging Models"
_NeurIPS.cc/2023/Conference — NeurIPS 2023 poster_

### Official Review · Reviewer_mRV1 · 2023-06-28

**Soundness:** 3 good
**Presentation:** 4 excellent
**Contribution:** 3 good
**Rating:** 6
**Confidence:** 3

**Summary:**

This paper identified two sources of performance degradation when merging fine-tuning models, (i) redundant parameter (ii) sing conflict and proposed TIES-MERGING to improve them.


**Strengths:**

1. The motivation to merge fine-gunning and performance is compelling.
2. Experiments were conducted with both NLP and vision, and the proposed method improved accuracy with both.
3. The proposed method is simple and computationally inexpensive, making it easy to reproduce.


**Weaknesses:**

1. There is no theoretical support for the proposed method.
2. In particular, the effect of sign conflict on performance is nontrivial and requires theoretical support.

**Questions:**

1. Is there any theoretical support regarding sign conflict?
2. I needed help understanding how to see Fig. 6. What is "Parameters With Nonzero Values" in Fig. 6.a mean?
3. In Figure 6, why does it look like you assume that a larger mean magnitude will result in better accuracy?
In other words, why do you know that a task vector with a small magnitude does not affect accuracy but also that a task vector with a large magnitude improves accuracy?

**Limitations:**

Yes.

---

> ### Author Rebuttal · Authors · 2023-08-10
>
>
> **Weakness 1, 2 and Question 1:** Theoretical support for the proposed method.
>
> **Answer:** Our work is supported by several past works that shed light on different components of our method. We discuss each of them below and will add a similar discussion to the updated paper.
>
> **Why Merging Works:** As mentioned in L91-102, model merging and weight interpolation work because models finetuned from the same pre-trained initialization demonstrate mode connectivity [1] which implies that the finetuned checkpoints lie in an approximately convex loss basin in parameter space [2]. The theoretical underpinnings of mode connectivity, therefore, support the merging procedure we develop.
>
> **Existence of Sparse subnetworks:** Past works [3,4] have shown that during model finetuning, most information is encoded in a small and sparse subnetwork inside the full model. Achille et al. [5] describe two phases of training where the first phase discovers the important connections and their topology between layers and the second phase fine-tunes this relatively fixed pattern. In our work, we find that the top-k% parameters form a subnetwork that is responsible for most of the performance and only considers merging them to avoid unnecessary interference from other parameters.
>
> **Relevance of Signs:** In addition, the Lottery Ticket Hypothesis [3] finds subnetworks (winning tickets) by magnitude pruning and shows that by re-training them with static sparsity starting from the initial weights, they reach similar or higher accuracy. However, they demonstrated that random initialization, with the same structure, does not suffice. This implies that the values of the weight are crucial for good performance.  Zhou et al. [4] build upon this and show that one may not need the exact weights at initialization to train these subnetworks (lottery tickets) but the signs may be sufficient. We also find that during merging signs are critical for model performance (see the main paper, Figure 7) and hence we explicitly focus on resolving the sign conflict when merging models.
> ___
> **Question 2:** I needed help understanding how to see Fig. 6. What is "Parameters With Nonzero Values" in Fig. 6.a mean?
>
> **Answer:** In Figure 2,  we provided the intuition of the different types of interference that exist when merging models and how it impacts the parameter values in the merged model. In Figure 6, we quantify the effect of both types of interference for real models that are being merged. As mentioned in L281-283, we bin the parameters of the model into three categories: (1) parameters that are redundant for all tasks (not in top-20% of any of the models); (2) parameters only influential for 1 model; and (3) parameters that are influential to more than one task. These categories help elucidate the impact of interference. For each category, we plot the means of the values in the merged model when using basic mean and trimming + disjoint mean. We know that zeroing out the redundant parameters does not affect the performance of the original task. However, removing redundant parameter values when merging models allows us to retain the change in that parameter's value introduced by another fine-tuned model. In contrast, taking the mean of all the values reduces the magnitude of the parameter in the merged model which might hurt the performance of a task that had a high magnitude in that direction. This is also explained in L49-59.
> ___
> **Question 3:** In Figure 6, why does it look like you assume that a larger mean magnitude will result in better accuracy? In other words, why do you know that a task vector with a small magnitude does not affect accuracy but also that a task vector with a large magnitude improves accuracy?
>
> **Answer:** We do not assume that a larger mean magnitude will result in better accuracy. However, in the trimmed task vector if the value of a parameter is high the parameter is influential for the task and we have observed that reducing the values of the parameters results in a degradation in performance. As noted in Figure 3, resetting the top magnitude values results in huge performance drops. Consequently, TIES merging removes all redundant parameters and only merges the influential ones (that lead to a drop in performance if reduced).
> ___
> [1] Linear Mode Connectivity and the Lottery Ticket Hypothesis
>
> [2] Loss surfaces, mode connectivity, and fast ensembling of dnns.
>
> [3] The Lottery Ticket Hypothesis: Finding Sparse, Trainable Neural Networks
>
> [4] Deconstructing Lottery Tickets: Zeros, Signs, and the Supermask
>
> [5] Critical Learning Periods in Deep Networks

---

> > ### Comment · Reviewer_mRV1 · 2023-08-14
> >
> > Thanks for the answer.
> >
> > I'm clear on Questions 2 and 3, but I need more theoretical backing for Question 1. I appreciate the experimental contributions of this paper. Given that there's a suggestion in common question 3 to add a suitable limitation, I am happy to raise my score.

---

> > > ### Author Response · Authors · 2023-08-14
> > >
> > > We are glad that we were able to answer all of your questions and that you have decided to update your score. And yes, we will add in our limitations section (as mentioned in the general response) that there is limited theoretical understanding on why model merging works. Thank you again for your time.

---

### Official Review · Reviewer_ioFQ · 2023-07-03

**Soundness:** 3 good
**Presentation:** 3 good
**Contribution:** 2 fair
**Rating:** 6
**Confidence:** 4

**Summary:**

The paper presents a novel method, TIES-MERGING, to merge models in the weight space for multitask learning. It observes an interference problem when linearly interpolating weights, and proposes a simple yet effective two-step solution: parameter trimming for small changes during fine-tuning and sign conflict resolution. The experiments in multitask shows that TIES improves performances, making it a notable (experimental) contribution to the literature of model merging.

**Strengths:**

* The paper's main strengths lie in its simplicity, as highlighted by its clear description and illustration.
* The paper successfully builds upon the "task arithmetic" task vector approach to report a new interference phenomenon, and then enhances performance with simple yet important modifications. Model merging is an important topic in multitask, and this paper fills a crucial gap in the current literature.
* The experimental framework is robust, with a focus on significant large-scale tasks across CV and NLP domains using recent architectures and fair benchmarks.

**Weaknesses:**

Despite its strengths, some areas require attention.
* The contributions, though valuable, are incremental, and the observed gains in multitask learning are consistent but arguably marginal, and the trimming/sign play only a marginal role in this gain.
* The experiments focus solely on models trained on different tasks. Yet, weight averaging is also useful to merge models trained on a single target task; on this model soups setup, I speculate that sign interference is less an issue, and that TIES would actually be detrimental as it would increase variance, thus limiting the benefits from combining multiple models, in particular under distribution shifts. As a minimum fix, the title should reflect this specificity, as the current one does not adequately reflect this scope limitation. A (naive) suggestion could be "Resolving Multitask Interference When Merging Models".
* Even whithin the multitask challenge, the experiments do not cover two important scenarios. First, multitask as better pretraining before fine-tuning on a target task (as in "Fusing finetuned models for better pretraining"). Second, multitask in the sequential patching scenario (as in "Patching open-vocabulary models by interpolating weight").
* Lack of analysis and clarity of the interpolating coefficient, for TIES and for the baselines. Specifically, without validation samples, could you clarify which values $\lambda$ is used: it seems that it's $\lambda=1/|num tasks|$ for weight averaging, $\lambda=0.4$ for task arithmetic (could you please point where you found this value), but $\lambda=1$ for TIES. Therefore, is the difference between task arithmetic/weight averaging in Table 1 simply due to the use of different $\lambda$? Thus (as further suggested from Table 3), scaling is an important factor, these different values of $\lambda$ hidden the true impact of your core contributions.
* Similarly, the ablation study in Table 3 could be made clearer. It is not clear whether the ablations are done one at a time or sequentially all together. If the latter is true, then why do we recover 65.5 on T5-base (when all 4 components are removed), while weight averaging performs 65.9? Moreover, what does it mean to remove "elect" while keeping "disjoint mean"? what does it mean to remove "scaling" (is it $\lambda=1/|num tasks|$ or something different)?

**Questions:**

* What is the impact of task similarity on the number of sign conflicts? Does the number of sign conflicts decrease when two models are fine-tuned on a shared task?
* Can you visualize the number of sign conflicts for different trimming ratios?
* Can TIES improves downstream transfer learning performances?
* Could you plot the curve performance while varying the coefficients given to different tasks: see Pareto curves in "Patching open-vocabulary models by interpolating weight", but also in "Pareto manifold learning: tackling multiple tasks via ensembles of single-task models", a missing yet important related work.
* Could you enrich the ablation study to further clarify/highlight the different impacts of the key contributions.
* How does ensembling of predictions behave in comparison with weight interpolation?

**Limitations:**

The authors have highlighted in Sections 7.3 and 7.4 that a key limitation is to accurately elect the sign.
Yet, the paper would benefit from a dedicated limitation section. It should ideally include that:
- weight interpolation lacks proper theoretical understanding,
- their approach is only (verified) for multitask learning,
- the averageability relies on a large pretraining and "good" hyperparams,
- TIES may be less impactful in a "sequential patching" scenario,
- they still lag behind the simultaneous multitask learning.

---

> ### Author Rebuttal · Authors · 2023-08-10
>
>
> **Weakenss 1:** The contributions, though valuable are incremental.
>
> **Answer:** We note that the improvements of TIES-Merging over Task Arithmetic are precisely due to the role of trimming values and electing signs when merging since there are no other differences between the methods. Moreover, the improvements reported in Table 1 (with validation) range from 0.7% to 3.6% over 11 tasks for IA3, 7 tasks for T5, and 8 tasks for vision experiments. Across all tasks that are averaged over, the improvement for IA3 is 27.5%, 4.9% for T5-base, 25.2% for T5-large, 14.4% for ViT-B/32, and 12% for ViT-L/14 across all tasks. Hence, we submit that these improvements are not marginal.
> ___
> **Weakness 2 (Part 1), Question 3, and 6:** "Model soups" experiments, merging multiple checkpoints from the same task, comparison with ensembling.
>
> **Answer:** We performed additional experiments to merge multiple checkpoints trained on the same task (as done in Model Soups) and also compared with ensembling. We use the experimental setting and code from "Merging Models with Fisher-Weighted Averaging", i.e. merging differently fine-tuned BERT models trained on RTE, MRPC, and WNLI. For each of these datasets, we select 10 fine-tuned checkpoints based on bert-base-uncased from the Hugging Face Hub and then merge them using different methods and report the results below.
>
> ||RTE|MRPC|WNLI|
> |-|-|-|-|
> |Averaging|59.93|78.19|56.34|
> |Fisher|65.7|81.37|52.11|
> |Ensemble|70.76|86.03|45.07|
> |Task Arithmetic|71.84|86.03|59.15|
> |TIES|72.2|86.76|58.75|
>
> From the results presented above (also presented as Table 1 in Rebuttal PDF), we observe that TIES merging works the best in all cases except WNLI, where it only slightly underperforms Task Vectors. Notably, TIES merging provides a dramatic boost over both Fisher Merging and averaging, and **outperforms ensembling in all cases**.
> ___
> **Weakness 2 (Part 2) and Question 1:** Sign interference is less of an issue for models trained on a single task.
>
> **Answer:** In rebuttal PDF Figure 2, we show that sign conflicts exist even within models trained on the same task. We plotted the sign conflict between the 10 checkpoints of RTE, MRPC, and WNLI from HF (mentioned above). As the number of checkpoints increases, sign conflict increases. We also compare this with the sign interference when merging different task checkpoints and find a similar degree of interference in all of these cases. Hence, sign conflicts exist even within models trained on the same dataset. We suspect that this is because models are highly overparameterized and hence there are multiple subnetworks (subsets of parameters) that could lead to the same performance where different finetuning runs update the same parameters in different directions.
> ___
> **Weakness 3:** multitask as better pretraining before fine-tuning on a target task.
>
> **Answer:** Based on your suggestion, we performed additional experiments following the setting from "Fusing Finetuned Models for Better Pretraining", specifically merging the same three GLUE tasks from the previous experiment (RTE, MRPC, WNLI). We take finetuned bert-base-uncased model checkpoints for 8 GLUE tasks (wnli, sst2, rte, qnli, mrpc, cola, mnli, qqp) from HF. When finetuning on a given task, we merge all the other seven checkpoints together (apart from the chosen task) and use that as an initialization for fine-tuning on the chosen task. The results are presented in rebuttal PDF Table 2 and copied below. We find that TIES merging works well in this setting and outperforms all other merging methods by a significant margin (apart from Averaging for WNLI).
>
> |Init Method|RTE|MRPC|WNLI|
> |-|-|-|-|
> |PTM Init|66.42|81.86|56.33|
> |Average|75.81|86.51|56.33|
> |Task Arithmetic|78.33|86.27|50.7|
> |TIES|80.14|87.99|54.92|
> ___
> **Weakness 4:** Lack of analysis and clarity of the interpolating coefficient….
>
> **Answer:** Please refer to our general response on how the hyperparameters were selected and their effect on performance. In addition, we note that scaling is indeed an important factor for both Task Vectors and TIES merging because after merging the scale needs to be recalibrated. However, in all the experiments (apart from Table 1, three rows without validation set) for both the methods we fairly tune the $\lambda$ as a hyperparameter and use the best value. Please refer to the general response for other details.
> ___
> **Weakness 5 and Question 5:** Similarly, the ablation study in Table 3 could be made clearer.
>
> **Answer:** As mentioned in L311-314, the ablations are done by removing one component at a time from the full TIES merging method. Removing elect while keeping the disjoint mean refers to taking the mean of values with signs +1 and -1 but not including the 0 values of the trimmed task vectors in the mean. Removing disjoint mean but trimming and electing refers to taking the mean of the values with the elected signs and the 0 for the trimmed values. Removing scaling means using $\lambda=1$. Removing components one at a time allows us to quantify the effect of each component. If you have any other suggestions for ablations, please feel free to let us know and we will add it to the paper.
> ___
> **Question 2:** Can you visualize the number of sign conflicts for different trimming ratios?
>
> **Answer:** In rebuttal PDF Figure 3, we merge the 10 bert-base-uncased models finetuned on for different glue tasks (RTE, MRPC, and WNLI) and plot the sign conflict as a function of K. As we keep more and more parameters, the sign conflict increases and reaches almost 80%. This is also expected as there are many more nonzero parameters that can create conflict even if their magnitude is small.
> ___
> **Question 4:** Plot the curve performance while varying the coefficients given to different tasks and missing important related work.
>
> **Answer:** We will cite this in the updated version! For the curves please refer to the general response Question 2.

---

> > ### Comment · Reviewer_ioFQ · 2023-08-10
> >
> > I would like to thank the authors for the clear rebuttal and the additional experiments that clarify the paper and its contributions. My main remaining concern is the small gains in Table 1 from the rebuttal wrt Task arithmetic; thus, I still think that TIES is useful in "multitask" setups, but perhaps less in a single-task/model-soups scenario. Yet, I am happy to increase my score to a 6.

---

> > > ### Author Response · Authors · 2023-08-10
> > >
> > > We are glad that we were able to address most of your concerns and that you have decided to update your score! However, a small note is that we are not able to see the updated score on our end yet, so it would be great if you can double-check once. Thanks!

---

### Official Review · Reviewer_b9mw · 2023-07-06

**Soundness:** 3 good
**Presentation:** 3 good
**Contribution:** 2 fair
**Rating:** 5
**Confidence:** 3

**Summary:**

This paper delves into the challenge of integrating multiple task-specific fine-tuned models into a singular, multitask model, without necessitating additional training. The authors identified that current methodologies overlook the interference that occurs between parameters of different models. This interference can be attributed to two primary sources: redundant parameter values and conflicts in the sign of a given parameter's values across various models. To counteract these issues, the authors introduced a novel method, TIES, which incorporates three key steps: (1) resetting parameters that underwent minimal changes during fine-tuning, (2) resolving conflicts in sign, and (3) merging only those parameters that align with the final agreed-upon sign. The proposed method demonstrated SOTA performance across different settings.

**Strengths:**

- The paper provides a comprehensive analysis of the sources of interference in existing model merging methods, specifically pinpointing redundancies in model parameters and disagreements between parameter signs. This thorough examination underpins the motivation for the proposed method.
- The proposed method demonstrates robust performance across a wide array of conditions, including various modalities, domains, task quantities, model sizes, architectures, and fine-tuning settings. This versatility underscores the method's adaptability and broad potential for application
- The informative and insightful Section 7, which delves into the significance of different components, particularly the estimation of correct signs during the merge process, serves as a valuable resource.

**Weaknesses:**

- The proposed method is quite heuristic and it would make the paper stronger if the authors can provide more theoretical analysis of the proposed method.
-  The author did not mention the limitations of their method and potential future work. The authors could further discuss them, which will provide a more balanced view of the method and give readers an idea of the potential directions for future research.
- The paper could be improved by discussing more real-world applications of the studied problem, integrating multiple task-specific fine-tuned models into a singular, multitask model, without necessitating additional training. Why is this problem important? This would help to demonstrate the practical value of the method and its potential impact in real-world scenarios.
-  It would be beneficial to include a sensitivity analysis of the hyperparameters used in the method. Although the authors provided a generic recipe of TIES with fixed hyperparameters, it would be good to see how sensitive the method is to different hyperparameters.


**Questions:**

I don't have further questions for now and I look forward to the authors' enhancements on the weaknesses mentioned above.

**Limitations:**

The author did not mention the limitations of their method and potential future work. I have listed some suggestions for improvement in the weaknesses section.

---

> ### Author Rebuttal · Authors · 2023-08-10
>
>
> **Weakness 1:** Theoretical Justification
>
> **Answer:** Our work is supported by several past works that shed light on different components of our method. We discuss each of them below and will add a similar discussion to the updated paper.
>
> **Why Merging Works:** As mentioned in L91-102, model merging and weight interpolation work because models finetuned from the same pre-trained initialization demonstrate mode connectivity [1] which implies that the finetuned checkpoints lie in an approximately convex loss basin in parameter space [2]. The theoretical underpinnings of mode connectivity, therefore, support the merging procedure we develop.
>
> **Existence of Sparse subnetworks:** Past works [3,4] have shown that during model finetuning, most information is encoded in a small and sparse subnetwork inside the full model. Achille et al. [5] describe two phases of training where the first phase discovers the important connections and their topology between layers and the second phase fine-tunes this relatively fixed pattern. In our work, we find that the top-k% parameters form a subnetwork that is responsible for most of the performance and only considers merging them to avoid unnecessary interference from other parameters.
>
> **Relevance of Signs:** In addition, the Lottery Ticket Hypothesis [3] finds subnetworks (winning tickets) by magnitude pruning and shows that by re-training them with static sparsity starting from the initial weights, they reach similar or higher accuracy. However, they demonstrated that random initialization, with the same structure, does not suffice. This implies that the values of the weight are crucial for good performance.  Zhou et al. [4] build upon this and show that one may not need the exact weights at initialization to train these subnetworks (lottery tickets) but the signs may be sufficient. We also find that during merging signs are critical for model performance (see the main paper, Figure 7) and hence we explicitly focus on resolving the sign conflict when merging models.
> ___
> **Weakness 2:** Limitation and Future Work
>
> **Answer:** Please refer to the general response Question 3.
> ___
> **Weakness 3:** Discussing more real-world applications of the studied problem. Why is this problem important?
>
> **Answer:** As mentioned in L24-28, there are thousands of FT checkpoints released on Huggingface Hub every day and model merging allows us to reuse these checkpoints to create other models with desired properties: (1) Merging provides a cheap way to create and obtain models with specific abilities by merging multiple models. (2) Merging can also be useful in settings where the user cannot explicitly share the data, hence the user cannot share the data but share the model weights which can then be combined from multiple users to create a better model. (3) A process of merging multiple models together also allows for the collaborative development of open-source models where models are created and patched with specific abilities.
> ___
> **Weakness 4:** Sensitivity analysis of the hyperparameters used in the method
>
> **Answer:** Please refer to our general response and Figure 1 in the Rebuttal PDF.
> ___
> [1] Linear Mode Connectivity and the Lottery Ticket Hypothesis
>
> [2] Loss surfaces, mode connectivity, and fast ensembling of dnns.
>
> [3] The Lottery Ticket Hypothesis: Finding Sparse, Trainable Neural Networks
>
> [4] Deconstructing Lottery Tickets: Zeros, Signs, and the Supermask
>
> [5] Critical Learning Periods in Deep Networks

---

> > ### Comment · Reviewer_b9mw · 2023-08-16
> >
> > I'm grateful for the authors' thorough responses and their initiative in addressing the highlighted concerns. I will retain my score.

---

### Official Review · Reviewer_Rfna · 2023-07-06

**Soundness:** 3 good
**Presentation:** 3 good
**Contribution:** 2 fair
**Rating:** 7
**Confidence:** 4

**Summary:**

Merge mutiple **fine-tuned** neural networks, each from an unique task (dataset), into one neural network by weight averaging. Denate the weights of pretrained network as $\theta_0$, fine-tuned network as $\theta_{\tau}$, the weight updating direction $\tau = \theta_{\tau} - \theta_0$

The paper introduces two tricks 1) for each $\tau$ only keep the top $k$% elements in $\tau$ according to their magnitude. (keep the others as zero), 2) during the weight averaging, only retain the dominate direction (+1 or -1) for each elements.

The experiments shows the improvements of the proposed Ties-Merging method over other weight averaging methods, such as naive weight averaging, task arithmetic, and regmean. But Ties-Merging is still significantly lagging behind MultiTask training.



**Strengths:**

- The writing is clear and easy to read.
- The introduced two tricks are resonable and easy to understand.
- Comparing with naive weight averaging, task arithmetic, Fisher averaging, and regmean, the proposed ties-merging performs very well.



**Weaknesses:**

- Even though the proposed Weight Averaging method out-perform other weight averaging methods, it still legs behind MultiTask training.
Furthermore, considering both the training cost and inference time, it is hard to see the benefit of the proposed method over mutitask training. As to other baseline methods, such as naive weight averaging, Fisher averaging and regmean, they are designed to solve IID (one fine-tuning task instead of mutiple different tasks) not OOD. These two facts reduce the strongness of the experiments.  (But it is fine. Because the two introduced tricks are good enough).








**Questions:**

- Simple averaging is $1/n \sum_{\tau} \theta_{\tau}  = \theta_0 + 1/n \sum_{\tau}\tau$
- Task Arithmetic  is $ \theta_0 + \lambda \sum_{\tau}\tau$

What is the parameter $\lambda$ of task arithmetic in Figure 5? If it is $0.4$ as line 210, why there is a big gap between simple averaging and task arithmetic in figure 5, ( given that $1/2=0.5, 1/3=0.33$)

**Limitations:**

no negative societal impact.

---

> ### Author Rebuttal · Authors · 2023-08-10
>
>
> **Weakness 1 (Part 1):** Method still legs behind MultiTask training.
>
> **Answer:** While we agree that the gap between TIES merging and multitask training is an important limitation, we note that multitask learning requires simultaneous training on all the training data at once. In contrast, when merging models, we only require access to individual-task models and do not require additional training. These individual task models can be trained asynchronously and recycled to create many different merged models, and are frequently shared e.g. on the Hugging Face Model Hub (where hundreds of thousands of fine-tuned models are available). Our methodological contribution identifies an important aspect of interference when merging task-specific models and shows notable improvements over existing methods.
> ___
> **Weakness 1 (Part 2):** The training cost and inference time make it hard to see the benefit of the proposed method over multitask training.
>
> **Answer:** We would like to clarify that our method does not require any additional training. In a real-world setting, most of these models would be trained by different people for their use cases and then uploaded to a common hub for public use. Moreover, the inference cost of the merged model is the same as that of the original model. Hence, there is no additional inference cost either.
> ___
> **Weakness 1 (Part 3):** Other baseline methods, such as naive weight averaging, Fisher averaging, and regmean, are designed to solve IID (one fine-tuning task instead of multiple different tasks), not OOD.
>
> **Answer:** RegMean was specifically designed for merging different tasks via their objective, and the RegMean paper includes experiments of merging different tasks in section 5.1. Similarly, weight averaging is also used to merge different tasks for better initialization [1]. Moreover, in PDF Table 1, we have provided additional experiments where we show that TIES merging outperforms other methods even when merging multiple checkpoints from the same task. Hence, even in the "IID" setting TIES outperforms other methods. For more details please refer to our response to reviewer ioFQ.
> ___
>
> **Question 1:** What is the of the parameter \lambda of task arithmetic in Figure 5? If it is 0.4 as line 210 …
>
> **Answer:** The value of $\lambda=0.4$ is only used for the experiments without a validation set in Table 1 (three rows); in all other experiments the value of $\lambda$ is tuned on a validation set. In Figure 5, for each subset of tasks that are merged, we select the best $\lambda$ on the validation set. Hence, it is not expected for averaging and Task Arithmetic to be close to each other and we see a difference in performances.
>
> [1] Fusing finetuned models for better pretraining, Choshen et. al. 2022

---

> > ### Comment · Reviewer_Rfna · 2023-08-10
> > **Response**
> >
> > Thanks for the answer.
> >
> >
> > **Rebuttal 1 & rebuttal 2**: individual-task models do not require additional training, since there are a lot shared models, e.g. Hugging Face.
> >
> > That is problematic in practice. These shared model may not be averageable, due to fine-tuning stragegies. To make two fine-tuned model be averageable, certain mild conditions need to be satisfied in practice, such as small learning rate, mild regularization, etc.  So individual-task models still require additional trainings with the proposed method.
> >
> > **Rebuttal 4**: the parameter  $\lambda$.
> > Could you provide the exact number of $\lambda$ of different "Number of Tasks" in Figure 5? So that readers and compare the number with "simple averaging" (green line).

---

> > > ### Author Response · Authors · 2023-08-10
> > >
> > > Thank you for your additional comments. Please find our responses below.
> > >
> > > **Rebuttal 1 & 2 Answers:** The ensembling/model soup experiments, included in our rebuttal merges random checkpoints taken from the huggingface hub that can potentially have different training schedules and unknown hyper-parameters. Similar to, section 3.1 of Fisher Merging [1], section 4.2, and Appendix D.6 of Task Arithmetic [2], we observe that merging models trained on checkpoints trained by the community satisfies the underlying merging criteria in most cases and leads to good performance. However, if there are cases where merging some checkpoint degrades the performance, we can use the Greedy Soup recipe from the Model Soups [3] to eliminate such problematic checkpoints during merging. Moreover, we agree that additional training might lead to even better-merged models. But were able to successfully merge and use models from the HF hub without additional training and this observation on merging checkpoints is in line with many successful past works like Model soups [3], Task Arithmetic [2], and Fisher Merging [1].
> > > ___
> > >
> > > **Rebuttal 4 Answers:** As mentioned in L265-267, For Figure 5, for each number of tasks we take at most 10 random subsets of the 8 tasks we were considering. The solid line is the avg of the merged performance of these different runs. Below we provide the optimal lambdas for the different subsets of tasks we merged for both TIES and Task Arithmetic, note that for averaging its always 1/(# tasks). Each entry in the list is the optimal lambda for a particular subset of tasks.
> > >
> > > TIES (2 tasks) -> [1.7, 1.9, 2, 2, 1.1, 1.5, 1.6, 1.8, 1.9, 1., 5]
> > >
> > > Task Arithmetic (2 tasks) -> [1, 0.9, 1, 1, 0.9, 1, 0.9, 0.9, 0.9, 1]
> > >
> > > TIES (3 tasks) -> [1.2, 2, 1.5, 1.9, 1.8, 1.7, 1.4, 2, 3, 1.9]
> > >
> > > Task Arithmetic (3 tasks) -> [1, 0.7, 0.7, 1, 1, 0.9, 0.7, 0.7, 0.9, 1]
> > >
> > > TIES (4 tasks) -> [1.5, 1.3, 1.3, 1.8, 2.3, 1.7, 1.8, 1.7, 1.9, 1.5]
> > >
> > > Task Arithmetic (4 tasks) -> [0.8, 0.7, 0.7, 0.7, 0.6, 0.7, 0.7, 0.8, 0.6, 0.7]
> > >
> > > TIES (5 tasks) -> [2, 2, 2, 1.8, 1.7, 2, 1.6, 2.1, 1.6, 1.3]
> > >
> > > Task Arithmetic (5 tasks) -> [0.7, 0.8, 0.6, 0.8, 0.7, 0.6, 0.6, 0.6, 0.6, 0.7]
> > >
> > > TIES (6 tasks) -> [1.6, 1.7, 1.7, 1.2, 1.7, 1.7, 1.5]
> > >
> > > Task Arithmetic (6 tasks) -> [0.6, 0.5, 0.5, 0.5, 0.7, 0.5, 0.6]
> > >
> > > TIES (7 tasks) -> [1.7]
> > >
> > > Task Arithmetic (7 tasks) -> [0.5]
> > >
> > > ___
> > > [1] Merging Models with Fisher-Weighted Averaging.
> > >
> > > [2] Editing Models With Task Arithmetic.
> > >
> > > [3] Modelsoups: Averaging Weights of Multiple Fine-Tuned Models  Improves Accuracy Without Increasing Inference Time.

---

> > > > ### Comment · Reviewer_Rfna · 2023-08-11
> > > > **Response**
> > > >
> > > > **Rebuttal 1 & 2 Answers** is good. I suggest to add this discussion in the final version: "Not all public avialiable checkpoints are avarageable. When the models are not averageable, one can eliminate them".
> > > >
> > > > The $\lambda$ numbers in **Rebuttal 4 Answers** is good. Now the readers know it is larger than 1/(# tasks) in general. And know what contributes the good performance compared with naive averaging. I suggest to add these numbers in the final version, e.g. appendix.
> > > >
> > > > Thanks for your answer.

---

> > > > > ### Author Response · Authors · 2023-08-11
> > > > >
> > > > > Thank you for your time, effort, and questions! We are glad we could answer your concerns. We will clarify these details further in the final paper.

---

### Official Review · Reviewer_rBCc · 2023-07-25

**Soundness:** 3 good
**Presentation:** 3 good
**Contribution:** 3 good
**Rating:** 6
**Confidence:** 4

**Summary:**

This paper proposes to resolve the interference of model merging, a solution to combine multiple task-specific models into a single multitask model. It demonstrates two major sources of interference, including redundant parameter values and sign conflict and proposes solutions to resolve the interference.

**Strengths:**

The paper demonstrates some interesting research insights and the solutions are simple and clear.

**Weaknesses:**

1: I question the necessity of Section 7.3.  According to the network pruning literature, a high magnitude is always an implication of importance/sensitivity, and a widely used pruning metric. Flipping the sign of or pruning away the Top-k% parameters can cause significant performance drop is well-known to the literature. It seems to have weak connection with this paper’s main argument.

2: I did not find the discussions regarding limitations and future work.

3: More analysis on the experimental results are required. For example, what are the effects of the proposed on each task respectively? Why do some tasks perform better than others?


**Questions:**

Refer to questions.

**Limitations:**

The authors do NOT adequately address the limitations.

---

> ### Author Rebuttal · Authors · 2023-08-10
>
> **Weakness 1:** Necessity of Section 7.3 and pruning away the Top-k% parameters can cause a significant performance drop is well-known in the literature.
>
> **Answer:** As mentioned in L166-168, we would like to clarify that we are pruning the task vectors (i.e. the difference between the fine-tuned model and the pre-trained model), not the model's parameters. Prior work in the pruning literature mostly focused on pruning model parameters. Furthermore, we provide experiments and analysis to provide more intuition and motivation behind pruning task vectors. For example, Figure 3 shows that deleting the bottom 80% of the values from the task vector doesn’t affect performance while Figure 7 shows that the signs of the remaining top 20% of the parameters in the task vector are very crucial.
> ___
> **Weakness 2:** Limitations and future works
>
> **Answer:** Please refer to our general response Question 3.
> ___
> **Weakness 3:** What are the effects of the proposed on each task respectively? Why do some tasks perform better than others?
>
> **Answer:** We provide per-task results in Appendix Tables 6-12. For the effect of merging on individual tasks, we hypothesize that the improvements over other methods are in cases where there is significant interference between tasks. However, when merging multiple tasks it is harder to predict the effect on individual tasks and would be a good research question for future study.

---

### Author Rebuttal · Authors · 2023-08-10

We thank all the reviewers for their time and for providing constructive comments for enhancing the paper. We appreciate that reviewers recognized:

- That our paper fills a crucial gap in the current literature (ioFQ).
- Our notable experimental contribution (reviewer ioFQ) via robust, comprehensive, and fair experimental setup (reviewer ioFQ, b9mw, Rfna).
- Simplicity (reviewer mRV1, ioFQ, Rfna, rBCc), clear description (reviewer ioFQ, Rfna), and computational efficiency (reviewer mRV1).
___
In the responses, we include the following additional experiments showing,

1. ("Model soups" experiments) TIES outperform other methods when merging checkpoints trained on the same task (Rebuttal PDF Table 1).
2. ("Fusing for better Finetuning" experiments) TIES provides a better initialization compared to other methods for fine-tuning (Rebuttal PDF Table 2).
3. Effect of hyperparameters on the method’s performance (Rebuttal PDF Figure 1).
4. A significant amount of sign conflicts exist even when merging different checkpoints of the same tasks (Rebuttal PDF Figure 2).
5. Sign conflict increases as we vary the top-k threshold (Rebuttal PDF Figure 3).
___

### **Common Questions.**

**Question 1:** Clarification on the chosen values of lambda for different methods.

**Answer:** As a first point of clarification, in all of our experiments apart from a subset of rows in Table1, $\lambda$ is a hyperparameter and we tune for it over the validation set. All the hyperparameters were chosen in this way for all the methods, including TIES merging and all baselines.

When there is no validation set available, it is not possible to tune hyperparameters in this way, so we use a fixed value of $\lambda = 1$ for TIES (L654-659 and L205-206), and $\lambda=0.4$ for task arithmetic (L210). Weight averaging does not consider $\lambda$ as a hyperparameter and always uses 1/(# tasks) with or without a validation set (L185-186). In Appendix D3 of the task arithmetic paper, they suggested that $\lambda$ values between 0.3-0.5 work best for most cases and we therefore used 0.4 (the midpoint) in our experiments without a validation set.

TIES merging takes the disjoint mean of the task vectors which already normalizes the values by the number of models with non-zero values for a parameter. Hence, we use $\lambda = 1$ for TIES. In contrast, Task Arithmetic adds all the task vectors  ($\theta_m = \theta_{init} + \lambda * \sum \tau_i$), and hence a $\lambda$ value of 1/(# tasks) for task arithmetic is in spirit similar to using $\lambda = 1$ for TIES.
___
**Question 2:** Effect of hyper-parameters \lambda and k on the performance.

**Answer:** In Rebuttal PDF, Figures 1 (left and middle), we plot the effect of $\lambda$ on the performance when merging T5-base and T5-large models trained on GLUE (Similar to Table-1). For TIES merging, we vary $\lambda$ around the value 1 because TIES takes the mean of task vectors, whereas task arithmetic adds up the task vectors. Hence, a $\lambda$ value of 1 for TIES is similar to using $\lambda$ = 1 / (# tasks) for Task Arithmetic. The range of 0.8-1.8 for TIES was selected based on preliminary experiments on the PEFT setting (as mentioned in L654-659 and L205-206).  We find that TIES-Merging is much less sensitive to changes in $\lambda$ (with an accuracy range of 68-75 across the considered values of \lambda) compared to Task Arithmetic (with an accuracy range of 55-75). For the effect of k ( Rebuttal PDF Figure 1, right), we increment the value of k in steps of 10 and skip k=0 as that would select no parameters. We observe that as K increases the performance drops and then saturates. However, we would like to note that this curve might change based on the distribution of the values in the task vector.
___
**Question 3:** Limitation and future works.

**Answer:** We will add the following discussion to the main paper in the revised version.

As pointed out by reviewer ioFQ, our works share the same general limitations of existing merging methods, like (1) a limited theoretical understanding of why and when weight interpolation works, what are the important underlying factors, and its proper connections with mode connectivity; (2) that merging relies on common initialization and model architecture; and (3) merging individual-task models to create a multitask still lags behind the simultaneous multitask training. Moreover, it is not clear how to select the checkpoints for merging in order to create multitask models useful for specific domains. In addition, while our method provides a way to choose signs when merging task vectors, we still find that using the signs from a multitask model performs better. Some potential future works include figuring out a good way to estimate multitask signs without having access to the multitask model as this has the potential to bridge the gap between multitask merging and multitask training (as demonstrated in Section 7.4).

---

### Decision · Program_Chairs · 2023-09-21

**Decision:**

Accept (poster)

**Comment:**

The paper addresses the task of dataless model merging where one needs to merge the weights of several models into a single multitask model without any data available. The paper presents a novel simple approach to this problem and shows empirically that in can help reduce the gap between fine-tuning and dataless merging. The paper is also easy to read and has a robust experimental framework. As such we believe this paper should be accepted as part of this conference.